# Characteristics of Harmful Text:
# Towards Rigorous Benchmarking of Language Models

**Maribeth Rauh**[*]    **John Mellor**    **Jonathan Uesato**    **Po-Sen Huang**    **Johannes Welbl**

**Laura Weidinger**    **Sumanth Dathathri**    **Amelia Glaese**    **Geoffrey Irving**

**Iason Gabriel**    **William Isaac**    **Lisa Anne Hendricks**

DeepMind

## Abstract

Large language models produce human-like text that drives a growing number of applications. However, recent literature and, increasingly, real world observations, have demonstrated that these models can generate language that is toxic, biased, untruthful or otherwise harmful. Though work to evaluate language model harms is under way, translating foresight about which harms may arise into rigorous benchmarks is not straightforward. To facilitate this translation, we outline six ways of characterizing harmful text which merit explicit consideration when designing new benchmarks. We then use these characteristics as a lens to identify trends and gaps in existing benchmarks. Finally, we apply them in a case study of the Perspective API, a toxicity classifier that is widely used in harm benchmarks. Our characteristics provide one piece of the bridge that translates between foresight and effective evaluation.

## 1  Introduction

Pretrained autoregressive English language models (LMs) like GPT-3 [21], Jurassic-1 [72], and Gopher [85] cover a vast space of possible use cases [19], from code generation to customer-service chat.[2] Text generated by LMs also has the potential to cause harm if models are not developed and deployed carefully. In light of this, many works have documented both existing and potential harms arising from generated text [11, 102, 62], ranging from misinformation [74] to the reinforcement of social biases through the perpetuation of stereotypes [95].

An emerging body of work is already dedicated to benchmarking LM harms (see Table 1). However, for many known or anticipated harms, current evaluation tools are imperfect [17, 106, 103, 95]. This is supported by the work analyzing the Gopher model [85], in which the authors observed a variety of shortcomings in benchmarks, such as unclear desiderata and poorly defined demographic groups.

Outside language modeling, the broader machine learning (ML) fairness community has documented sociotechnical[3] insights that can help bridge the gap between foresight and evaluation, drawing on domains including medical applications [80], facial recognition [22], and recommender systems [41].

---

[*]Corresponding author: mbrauh@deepmind.com

[2]We refer to English language models as language models as all models and benchmarks in this study are in English.

[3]A term describing "systems that consist of a combination of technical and social components" [92].

36th Conference on Neural Information Processing Systems (NeurIPS 2022) Track on Datasets and Benchmarks.

For example, ML fairness research has established the importance of social context in determining what the benefits and risks of a technology will be in practice [60, 77, 92], suggesting this area needs to be explicitly considered in the evaluation of LMs, as well.

Drawing on existing critiques, our own experience analyzing Gopher [85], and lessons from the broader ML fairness community, we identified characteristics (section 2) of harmful text which have implications for benchmark design. From a set of potential characteristics, we selected (1) harm definition; (2) representational harm, allocational harm, and capability fairness; (3) instance and distributional harm; (4) context; (5) harm recipient; and (6) demographics affected.

Our characteristics support benchmark design in multiple ways. First, by mapping existing benchmarks onto the characteristics (subsection 3.1), we establish a shared vocabulary and identify gaps in current benchmarks. For example, we reveal a lack of benchmarks considering harmful language in longer textual contexts. A single benchmark cannot cover all harms, but our characteristics allow explicit understanding of what a benchmark might (and might not) capture. Second, the characteristics enable us to analyze whether these benchmarks measure what they claim to (subsection 3.2). As a case study, we apply our characteristics to the Perspective API[4], a toxicity classifier widely used in LM harm benchmarks. We observe, for example, that our "harm recipient" characteristic illuminates a potential mismatch between the API's design and how it is used to measure LM harms. Finally, we believe our characteristics can be used to guide the design of more rigorous benchmarks. Each characteristic makes key design decisions explicit, helping to avoid common pitfalls. We hope that our analysis and proposed characteristics will sharpen future benchmarks tackling LM harms.

## 2 Harm Characteristics

Our development of the characteristics was driven by considerations relevant to benchmark design and what we believe would be most useful for concrete next steps in that space. From a set of candidate characteristics (see Appendix D), we selected a subset using the following criteria: applicable across a variety of harms; relevant to, but not always discussed in, existing benchmarks of LMs; most useful for avoiding common benchmarking design pitfalls; and minimal overlap with other characteristics. Following a description of each characteristic, we include questions it may raise during benchmark design in order to concretize the characteristic.

### 2.1 Harm Definition

> **Harm**: The real world effect on people that the evaluation's metrics aim to approximate.

Existing work has provided an overview of the potential risks from LMs [102, 11], and existing benchmarks usually start with a harm definition, e.g., "anti-Muslim bias" [5]. However, these are sometimes under-specified [16] and might be dependent on other characteristics (e.g., demographic groups and application contexts). As opposed to relying on predefined definitions of harms like "bias" or "toxicity", we encourage practitioners to specify what these terms mean in the context of their benchmarks. Additionally, there can be an unintentional shift between how the harm is defined and what is measured in practice. Initially, the selected definition guides a benchmark designer's responses to the questions that each of the following characteristics raises. Then, as each is considered, they enable further refinement of what exact definition of harm a benchmark aims to measure. By doing so, the shift between definition and what was encoded will be avoided or occur intentionally.

**Example Questions.** Where does the benchmark designers' concept of harm originate, and does it have a particular context or legacy, e.g., in literature, industry, practitioners' own lived experience? What does the harm include, and what is out of scope? What metrics best approximate this? If the harm definition is broad, how will the different ways it manifests be covered?

---

[4]https://www.perspectiveapi.com/

## 2.2 Representation, Allocation, and Capability

> **Representational harm**: When someone is represented or referred to in a negative, stereotypical, denigrating, or unfair way on the basis of their identity.
> **Allocational harm**: When resources, opportunities, or services are distributed in an inequitable way.
> **Capability fairness**: When LM performance is equal, or justifiably different, across groups.

The distinction between representational and allocational harm has been outlined in prior work [16, 30, 8], in reference to fairness-related harms. **Allocational harm** refers to the inequitable distribution of resources or opportunities, such as loans or jobs. This emphasizes a real-world outcome. Although **representational harms** are often upstream from allocational harms [30], representational harms can be damaging in their own right. For example, Collins [28] shows how stereotypes can serve as "controlling images," a justification for oppression.

Real-world disparities that are a result of LM-generated text are rarely benchmarked. Thus, we extend this taxonomy to include **capability fairness**, which measures performance disparity on any task. Frequently, metrics are a proxy for the benefits a system's creators expect it to bring, and there is capability unfairness if these benefits accrue in an inequitable way. For example, if a system answers questions about one group less accurately than another group (as is done in [43]), this is a capability fairness issue. Although such a benchmark is abstracted from a real-world outcome, in practice they are easier to create, and we might expect differences in performance to translate into subsequent downstream harms.

**Example Questions.** What is the relationship between what is measured and real-world harm? How is this harm, and the performance on associated metrics, likely to be distributed across groups?

## 2.3 Instance and Distributional

> **Instance harm**: A single LM output or interaction which is harmful by itself.
> **Distributional harm**: LM outputs or interactions which are harmful in aggregate.

An **instance harm** is caused by a single LM output or interaction. If a language model outputs a slur, the potential for harm can be defined by reference to that specific output. In contrast, **distributional harms** are observed over many independent model outputs and can only be measured by observing the model's aggregated behavior. For example, outputting *"he was a doctor"* is not harmful in itself, but if the model refers to doctors as male more often than any other gender (when not specified by the context), this could constitute a distributional harm. This distinction is similar to Khalifa et al. [64]'s "pointwise" and "distributional" constraints and is also referenced in analysis of Gopher [85] and PaLM [26] outputs.

This distinction is particularly useful when formulating metrics and desired system behavior. For an instance harm, it may make sense to aim for an overall reduction in the type of harmful output (e.g., slurs). However, when measuring distributional harm, the metrics are often comparisons of a key metric (e.g., average sentiment score) between groups.

**Example Questions.** Does the type (instance or distributional) the metric captures match the type implicit in the initial harm definition? If a dataset includes both types, how does this impact metrics?

## 2.4 Context: Textual, Application, Social

> **Textual context**: The length of the text being evaluated and of content it is conditioned on, such as a prompt.
> **Application context**: What the LM is being used for and how it is deployed. This includes user experience and the software system in which it is embedded.
> **Social context**: Culture, geography, history, as well as users' attributes, e.g., language or technological fluency.

Recent language models have the capacity to make use of long range **textual context** [32], meaning they are often used for generating samples conditioned on long inputs. When harm benchmark metrics are calculated on unconditioned, sentence-length text, this does not account for the way a preceding conversation, prompt, or other text input may affect the harmfulness of the output at hand. For example, *"I launched experiments with a bug in them. They should all be killed."* might not be considered toxic. However, the second sentence on its own (*"They should all be killed."*) would likely be considered toxic. Both the length of text being evaluated and what that text is conditioned on may help reduce the ambiguity of a harm's presence in the text as well as capture a variety of situations in which the harm could occur.

**Application context** also informs what kind of outputs are inappropriate, undesirable, or harmful. What may be acceptable to output as part of summarizing a news article may be inappropriate in a customer service chat. Language models may even be used as a foundation for derivative tasks, such as the base of a classifier, for which knowledge of harmful language may be critical for performance [85]. As characterizing harmful outputs is challenging without an application in mind, we recommend practitioners explicitly consider in which cases their benchmark may or may not be relevant.

Finally, every application is shaped by a **social context** [60, 77, 92, 53], which includes a range of factors such as language and cultural norms in the community using a system. Harm definitions, in particular, tend to implicitly encode cultural norms, not only through the initial definition but also from different steps in the benchmark creation. This includes the values of annotators, the sources of annotated text (e.g., news sources), and the use of pre-made classifiers such as toxicity classifiers (see subsection 3.2). It is also important to consider which subsets of data may be "missing" because they are difficult to collect based on factors that vary by geography, such as internet access.

**Example Questions.** How much would additional text reduce ambiguity about the harm's occurrence? Is a harmful output benign in other applications? In what linguistic, geographical, and cultural context was the data collected? What aspects of the harm might be culturally dependent?

## 2.5 Harm Recipient: Subject, Reader, Author, and Society

> **Subject or topic**: The groups or individuals that the output contains reference to, directly or implicitly.
> **Reader**: Whoever reads the LM outputs.
> **Author**: The groups or individuals that an LM output could appear to be written by, e.g., if the LM outputs text claiming to be a woman or impersonating a specific person.
> **Society**: When no one is referenced but harm occurs widely, e.g., if an LM were used for weapons research.

When an individual or group of people are the **subject** of a model output, they can be harmed, regardless of if they ever interact with the system. For example, outputting an explicit stereotype may negatively impact a particular group, even if members of that group never read the output text.

The **reader** is anyone who consumes the model's output. This may be an individual person, as in a one-to-one interaction with the model, or it may be many people, as when a model output is widely disseminated. Toxicity work that focuses on personal attacks exemplifies how harm can occur to a reader. Capturing such harms is challenging since a given output may not be harmful to all readers but the attributes of the reader are usually unknown at evaluation time.

LMs can operate as an "**author**" which represents a person or a group by using the first person, outputting text on behalf of someone (e.g., email autocomplete) or presenting a persona (e.g., as digital assistants). If a model with a persona claims a particular identity, the model could misrepresent or denigrate that identity group by perpetuating stereotypes, e.g., subservient digital assistants that have female personas [24]. Some applications use a language model to help a person communicate, such as automatic text completion in e-mails, creative applications, and machine translation. These uses could be harmful if text completions misrepresent user intentions (e.g., when *AI Dungeon* inserted sexually explicit content into users' stories [96]) or if a mistake in translation incorrectly attributes harmful language to a human speaker (e.g., [12]).

Many LM harms could have ramifications for **society** in general. However, current LM benchmarks typically quantify only narrow characteristics of text, e.g., "does this output espouse a conspiracy

theory?". While this may approximate complex, real-world harms, like whether LM-generated conspiracy theories undermine democracy, it does not measure such harms.

**Example Questions.** Is the harm primarily experienced by someone interacting directly with the LM or could it be problematic for someone not present? If the harm impacts a reader, author, or society, who does the benchmark assume the readers, authors, or relevant society are?

## 2.6 Demographic Groups

> **Demographics**: Subsets of the population, grouped according to aspects of identity, e.g., gender or ability. In practice, classification of group membership is not well defined because even a single facet of identity can be fluid, composed of differing and competing factors, or unobserved or incorrectly reported in data [88, 99].

Classical fairness metrics [25, 29, 75] usually require specifying a protected attribute, such as sexual orientation or race.The ML fairness literature has already begun grappling with the complexities of defining and selecting demographic groups. For more widely studied identities, many works have outlined pitfalls in current work and suggested how to move forward, such as Hanna et al. [48]'s discussion of the contextual and constructed nature of race and Keyes et al. [63]'s work demonstrating the need to move beyond binary gender classification. Meanwhile, many facets of identity are understudied in ML fairness literature, such as disability [55, 45], intersectionality, and legally protected characteristics beyond those defined in the United States [88, 89].

Here, we outline considerations specific to language data. First, relevant demographic groups might be challenging to identify from text. In the case of gender, benchmarks that rely on pronouns will only capture the identity of people discussed in the text and cannot evaluate harms to a reader. Both classifiers and lists of identity terms have been used to detect if text is about or addressed to a certain group [14],[5] but certain identity terms are difficult to detect without sufficient textual context. For example, **coded terms**, or dog whistles,[6] refer to groups in ways that are invisible to explicit term lists but problematic nonetheless. Offensive identity terms can also have homonyms with no identity connotation at all, such as the term "redskin" in the context of potatoes.

The concept of "**marking**" in sociolinguistics describes how minorities or under-priviledged groups are more likely to be "marked," or explicitly specified, e.g., "the gay man," while not specifying at all, e.g., "the man," will be assumed to refer to a man with the majority attribute (e.g., straight) [101]. Certain methods for measuring bias do so by substituting different identity terms and observing how the chosen metric varies. For such metrics, the concept of markedness has bearing on the results.

To compare a metric between groups, practitioners need to think carefully about which groups are compared against each other. These **comparison classes** should reflect historical and contemporary power dynamics between groups in a meaningful way. Getting this right means reasoning about the social context, and associated power structures, the benchmark and model are developed within. For example, when measuring stereotypes, text that negates a stereotype (*"Black people **will / won't** steal anything"*) is different from that which switches the group identifier (*"Mike was **poor / rich** and thought growing up in the projects was tough."*) [17]. This is an especially relevant in benchmarks which use sentence templates or pairs.

The prior points apply when a demographic group is the subject or reader of the output. However, when a model is given a persona, the dialect of a particular social group, i.e. **sociolect**, rather than pronouns or group labels are the natural unit of analysis. It is important to think about how to handle potentially harmful text based on its author(s) because, for example, terms that are slurs in one context may be reclaimed for in-group usage in others. When studying model outputs, the model is never an in-group speaker. However, if a benchmark labels all training documents that contain a reclaimed slur as harmful, it is likely to reduce performance on co-occurring language from the marginalized group.

**Example Questions.** How can the relevant demographics be referred to in text, and do these have connotations? Does the usage of these terms vary based on who uses them? If a benchmark compares

---

[5]An example of a widely used term list which includes many identity-related terms is the List of Dirty, Naughty, Obscene, and Otherwise Bad Words [1]

[6]For example, the use of the phrase "international bankers" to allude to anti-Semitic conspiracy theories [81]

| Benchmarks | Representational (R), Capability (C), or Allocational (A) | Distributional (D) or Instance (I) | Context | Subject (S) Reader (R) or Author (A) |
|---|---|---|---|---|
| RTP [40] | R | I | Sentences from web | S/R/A |
| TwitterAAE [13] | C | D | Tweets | A |
| SAE/AAVE Pairs [44] | R/C | D | Tweets; application agnostic | A |
| Winogender [87] | C | D | Coreference sents. by practitioners | S |
| Winobias [108] | C | D | Crowd sourced coreference sents. | S |
| Gender & Occ. [21, 85] | R | D | Sentences; prompts by practitioners | S |
| Deconfounding [43] | C | D | Crowd sourced QA | S |
| TruthfulQA [74] | n/a | I | QA written by practitioners | R |
| DTC [71] | R | D | Sentences from web | S |
| Muslim Bias [5] | R | D | Paragraph written by practitioners | S |
| BAD [107] | R | I | Crowd sourced chat bot dialogues | S/R |
| BOLD [37] | R | D | Sentences from Wikipedia | S |
| Stereoset [78] | R | D | Crowd sourced sentence pairs | S |
| Sentiment Bias [54, 21, 85] | R | D | Sentences; prompts by practitioners | S |
| BBQ [83] | C | D | QA written by practitioners | S |
| UnQover [70] | C | D | QA written by practitioners | S |
| PALMS [97] | R | I | QA written by practitioners | S/R |

Table 1: **Characteristics for different benchmarks.** We observe limited coverage for some characteristics: only four benchmarks consider instance harms, textual context tends to be short, and the subject is the recipient of harm in all but three benchmarks. See Appendix A for harm definitions, more detailed context, and demographics.

similar text with different demographic terms, which comparisons capture the structures of power and privilege underlying the harm?

# 3 Operationalizing the Characteristics

To make the characteristics concrete, we ground them in current benchmarks. First, we map a range of existing benchmarks used to measure LM harms onto our characteristics. We then use a case study of a widely used toxicity classifier, the Perspective API[7], to further illustrate how the characteristics can be used to make implicit design decisions explicit.

## 3.1 Mapping Existing LM Benchmarks

Mapping benchmarks onto the characteristics highlights potential gaps and strong trends in the benchmarking landscape (see Appendix A for a complete mapping). In particular, existing benchmarks measure distributional harms, short textual contexts, and cases where the harm recipient is the subject.

We focus on benchmarks that test if autoregressive LMs (as opposed to masked language models like BERT [36]) generate harmful outputs. All benchmarks consist of a dataset of text samples which are input to a model and an evaluation protocol to score the outputs. Metrics can either operate over sampled text, e.g., measuring the toxicity of sampled text, or assigned probabilities from the language model, e.g., computing the perplexity of text. We include benchmarks which test for harmful outputs on tasks which have been tackled by LMs in a zero-shot setting, such as question answering (QA).

**Harm Definition.** Benchmarks cover a wide range of harms, and we cover their definitions in detail as well as how we characterized each benchmark in Appendix A.

**Representation, Allocation, Capability.** No benchmarks directly measure inequitable allocation of resources or opportunities, but rather consider intermediate tasks. Hence, though we mark some benchmarks as measuring representational harm or capability fairness, we do not mark any as measuring allocational harm. Moreover, all benchmarks are still far from deployed use cases. Though some work has studied how bias propagates downstream through language technologies [42, 59], an open challenge in designing benchmarks for language model harms is better understanding which metrics reflect harms in deployed use cases.

Analyzing representational harms, allocational harms, and capability fairness require comparing representations or performance across groups. Some benchmarks, like TruthfulQA [74], which aims to measure disinformation, do not include group-based metrics. Though studying disinformation is worthwhile without group-based analysis, a group-based analysis could be informative (e.g., is

---

[7]https://www.perspectiveapi.com/

the model more untruthful when discussing particular groups?). We hope that by using the lens of "representation, allocation and capability" when creating benchmarks, practitioners can intentionally decide whether group-based analysis is useful for meaningful progress on the harm they are studying.

**Instance and Distributional.** Most harms are classified as distributional. However, sometimes benchmarks which intend to measure distributional harms inadvertently include instance harms in their dataset. For example, Stereoset [78] measures a distributional harm as the probability of the stereotype text and anti-stereotype text are compared. However, as noted in Blodgett et al. [17], some stereotypes are harmful and should not be output at all, regardless of the paired anti-stereotype's relative likelihood. Considering if harms are instance or distributional allows practitioners to ensure both datasets and metrics are aligned to measure the harm as intended.

**Context.** Examining textual context, we note that many benchmarks operate over short lengths of text. Furthermore, in Table 1, many application contexts are unspecified because benchmarks are applied on raw LMs without any particular application in mind.

Many datasets include samples written by practitioners, either by hand or with sentence templates. Though this allows for exact control by practitioners, datasets are likely to reflect practitioners' assumptions about social context. In BBQ [83], questions are written by the dataset creators, but they account for this by linking each bias tested to an external source. This documents the social context in which biases arise and might be considered harmful.

Language and dialect are important aspects of social context. We note all benchmarks in Table 1 are designed to measure harms in English, indicating a lack of linguistic and cultural diversity that is well documented across other language tasks [51, 9, 10, 23]. Analogous benchmarks in other languages might be challenging to create because existing measurement tools, like toxicity classifiers, do not work well in all languages [69], cultural norms might not transfer [88], assumptions in benchmark design might not translate,[8] and there may be fewer qualified native speakers on common annotation platforms. Though challenging, we believe building benchmarks in non-English languages is essential work and hope to see more benchmarks in other languages in the future.

**Harm Recipient.** In Table 1 we observe that some benchmarks assume a language model can have multiple roles. For example, RealToxicityPrompts [40] includes prompts which use the pronoun "I" ("persona"), "you" ("reader") and third person pronouns ("subject"). Overall, benchmarks most often measure when language model outputs harm the subject of the generated language.

TwitterAAE [15] and SAE/AAVE Pairs [44] explicitly measure the ability of models to generate language which aligns with a certain dialect, which could be seen as taking on a "persona" of someone who speaks a dialect. However, for many applications, the ability of the model to understand a user's dialect, as opposed to dialect generation, is important. If dialect generation correlates with dialect understanding, performance on TwitterAAE and SAE/AAVE pairs may approximate reader harm, e.g., if the LM works poorly for those using that dialect. By considering benchmarks through the lens of harm recipient, practitioners can be more explicit about differences in what benchmarks measure and potential real-world harms.

**Demographic Groups.** Current benchmarks consider a variety of demographic groups, which we catalogue in Table 3. For the benchmarks we include, gender is the most frequently studied. Race, religion and profession are also common. Sexual orientation, socioeconomic status, and intersectional biases are less well represented, perhaps in part because they are "unobservable" [99]. Which groups should be analyzed is application dependent [48] but, as practitioners may not have a specific deployment scenario in mind, it is worth discussing why particular groups and attributes are chosen for analysis, and the implications for interpreting results.

Seemingly minor choices in which demographic terms are chosen can impact analysis. In the Gender & Occupation evaluation in Rae et al. [85], we found that gender bias in LMs varies between gender terms like "female" vs. "girl." Additionally, majority or higher-status attributes are often not explicitly stated, or marked [101], in text. Both Rae et al. [85] and Blodgett et al. [17] outline how markedness influences analysis in Sentiment Bias and Stereoset [78]. Markedness is also relevant when comparing language mentioning marginalized groups to language mentioning majority groups, as is often done in distributional bias benchmarks [35]. For example, comparing the likelihood of models generating

---

[8]For example, see the discussion of creating Spanish WEAT in [42]

the bigrams "gay marriage" and "straight marriage" might not be meaningful as text rarely specifies marriage as "straight."

## 3.2 Case Study: the Perspective API in LM Benchmarking

To further demonstrate how the characteristics can be used, we conduct an in depth case study of a toxicity classifier, the Perspective API. Although not a benchmark itself, the Perspective API is an important building block of numerous LM harm benchmarks [103, 106, 107, 97, 90, 65, 33]. Using our characteristics as a lens, we can make design decisions explicit and enable their interrogation. In doing so, we observe how the characteristics highlight potential pitfalls.We include only the characteristics that are most insightful for analyzing the API; the rest are in Appendix C.

**Harm Definition.** Toxicity is a concept that originated in the field of content moderation, specifically of online social media platforms and news comment sections [91]. It emerged from work on online hate speech, and the term became widely used following the release of the Perspective API [104, 58]. The Perspective API defines toxicity as "a rude, disrespectful, or unreasonable comment that is likely to make someone leave a discussion." This definition is operationalized by asking humans to annotate if a given text is toxic [6]. Toxicity is intended to cover content ranging from sexually explicit to violent, posing a challenge for coverage.

*In LM Benchmarks:* This definition is used as-is because practitioners cannot modify the way toxicity is defined by the API.

**Context.** The Perspective API is trained with online comments drawn from sources including the New York Times (NYT) and Wikipedia, which encode a multitude of social contexts such as language and commenters' political views [4]. Social context is also encoded by the annotators, whose labels are based on their personal reactions to them. In terms of textual context, the comments were written in the context of the surrounding media, e.g., a news article or comment thread, though the toxicity classifier does not use this context when classifying text [105]. The intended applications [2] are "human assisted moderation," "author feedback," and better organization of comments.

*In LM Benchmarks:* LM harms need to be measured in a large and evolving set of applications [102]. Some applications may even benefit from a "toxic" LM, such as building a new toxicity classifier [90, 85, 49]. Even if an LM application aligns with that of the Perspective API, there remain differences in the textual and social context of each. For example, [85] reported that the Books slice of their in-house MassiveText dataset has a higher average toxicity than slices we expect to be more similar to the Perspective API training data, like News or Wikipedia. It is unlikely that the Perspective API would provide meaningful toxicity scores for generated language which differs substantially, e.g., in length, topic, style. For example, if the API over indexes on a specific word, would long LM samples be scored as toxic even though, in the full textual context, the word was not used in a toxic way?

Using a pre-trained classifier means the context of its training data, such as human annotations, will be transferred to the LM evaluation. Though it may still be a useful starting point, awareness of the difference in textual, application, and social context enables appropriately caveating results or developing complimentary benchmarks.

**Harm Recipient.** The Perspective API focuses on harm done to readers who may "leave a discussion" and, in effect, have their voices silenced [57]. When used for content moderation of human language, the author of the comment may also be harmed if their content is incorrectly flagged as toxic.

*In LM Benchmarks:* It may seem intuitive that what is permissible for humans to say is permissible for a model, but reader harm depends on their perception of who, or what, they are interacting with. What norms apply to LMs has not yet been widely established, and users may have different expectations of and reactions to model outputs if they understand that they come from a model [46]. A reader's perception of the characteristics and intention of the author affects how the reader interprets the text. For example, in-group usage of reclaimed slurs can be considered acceptable depending on who uses them [31]. However, even if an LM claims to be part of a group, it is not clear if users would find its use of reclaimed terms acceptable, as the model cannot actually be in-group. Moreover, the trade-offs which the Perspective API must navigate based on protecting the freedom of human speech is not a protection that applies to LMs. Finally, many LM benchmarks focus on if the subject of the text is harmed, which does not align with how the Perspective API was trained.

**Conclusions.** Through the lens of our characteristics, and complimented by empirical evidence seen in [103, 106], we observe where using the Perspective API in LM benchmarks faces challenges. The characteristics specifically highlight the mismatched context and the divergence between norms for human language and those emerging for machine language. It is common practice for classifiers of all kinds to be re-purposed far beyond their original contexts because building high quality datasets is challenging as well as under-valued [56]. Selbst et al. [92] refer to this as the portability trap, a "failure to understand how re-purposing algorithmic solutions designed for one social context may be misleading, inaccurate, or otherwise do harm when applied to a different context." The Perspective API's own model card explicitly states that automated moderation is a "use to avoid" [76, 2].

As a socially constructed concept, we encourage practitioners to develop and operationalize a definition of toxicity, informed by consideration of our characteristics, which fits the context and norms of their setting. For example, the concept of toxicity could be refined by asking "toxic according to who?" as suggested by the "Harm Recipient" and "Demographics" characteristics. Such analysis will sharpen future benchmarks tackling the important harms related to violent, hateful, abusive, and otherwise offensive language.

## 4 Discussion

**Related work.** Numerous works have surveyed the landscape of potential language model harms, both broadly [11, 102] and specific to social bias [52, 93, 95, 35, 61]. These surveys focus on identifying and defining language model harms; our work is complementary in that we point out other characteristics important for measuring language model harms. Some of our characteristics expand on the critiques in [17, 53], in particular our characteristics of context, recipient of harm, and representational versus allocational harm. We emphasize a sociotechnical analyses of language which we believe can be used alongside other proposed methods for reliability testing for language technologies [98, 86]. Finally, the dimensions of harmful text defined in [34] overlap with ours, but their focus is on harm to those involved in the research process itself.

**Limitations.** We chose to limit our work to the characteristics that we believe are applicable to a diversity of harms, are useful for analysis of existing benchmarks and common pitfalls, and therefore facilitate concrete next steps for benchmark design. Examples of characteristics we did not include are frequency, severity, covertness, and temporality. We expand on why these were not selected in Appendix D, and we leave such considerations to future work.

We note that these characteristics are imperfect abstractions. Some will apply more cleanly to certain types of harm while others may be less relevant. Their relationship to each other is also not entirely independent. Certain distinctions in one characteristic will frequently occur with another. Rather than a mandatory checklist, our goal is to provide a set of key considerations for reflection that will inevitably need tailoring across the diversity of language model harms and applications areas, and will need updating as both proliferate in the real-world.

Finally, our characteristics are designed specifically to analyze language output by LMs. In particular, we do not consider harms to annotators or practitioners in the development of benchmarks. Though our characteristics could be repurposed to study such harms, we believe that such harms deserve special consideration and point to [34] as promising work in this direction. Additionally, we do not consider how to characterize training datasets (see [39] for one example in this direction). It is possible our characteristics could be repurposed, and we would encourage more thought in this direction.

**Conclusions.** Translating anticipated risks into rigorous benchmarks is challenging. Drawing on existing critiques of language model harm benchmarks and insights from machine learning fairness research, we propose six characteristics to guide reflection and help pracitioners avoid common pitfalls when designing benchmarks.

We encourage practitioners to use these characteristics as part of an iterative process, in which they revisit what they set out to measure in relation to what they implemented. This enables practitioners to make the adjustments necessary to align their harm definition and what the benchmark measures in practice. Our analysis of porting the Perspective API to language model harm benchmarks highlights how difficult such alignment can be, and the issues that arise when they remain unaddressed. We also

encourage practitioners to include those with expertise beyond the field of machine learning, both in the form of other disciplines and through lived experience, when evaluating language model harms.

For several characteristics - instance and distributional harm, context, demographic groups, and harm recipient - we observe limited coverage in current benchmarks. The space of potential language model harms we can evaluate is huge, and existing work only covers a fraction of this space. It is unlikely one benchmark will capture everything, but our characteristics clarify gaps remaining in the benchmarking landscape. Building adequate benchmarks that touch on all characteristics poses a large challenge to the field.

In addition to guiding more rigrous benchmark design, we hope others will extend and refine these characteristics as our understanding of language model risks evolves. By synthesizing existing critiques of benchmarks and taxonomies of harm, we believe our proposed characteristics provide a constructive starting point to facilitate the translation of anticipated risks into safer and more beneficial language models.

## Acknowledgments and Disclosure of Funding

The authors received no specific funding for this work. We would like to thank Nat McAleese, Laura Rimell, Susannah Young, Edgar Duñez-Guzmàn, Suzanne Sadedin, Soham De, Stevie Bergman, Martin Chadwick, Ben Coppin, and Lucas Smaira for valuable discussion and feedback. In addition to helpful comments, we would like to give special thanks to Shakir Mohamed for encouraging and fostering this work.

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
