## Appendix Overview

Our appendix includes:

- **A**. Further details on our benchmark mapping included in subsection 3.1 of the main paper: our methodology in choosing benchmarks, descriptions of each benchmark, and a table outlining which demographic attributes are considered by each benchmark.

- **B**. A description of how we applied our characteristics to each benchmark during our mapping.

- **C**. Application of characteristics to the Perspective API, for those not included in the main content.

- **D**. Further details about our characteristic selection criteria and those omitted.

## A    Mapping Existing LM Benchmarks

Here we include details about our benchmark mapping analysis. Table 2 summarizes the inputs, outputs, and metrics used in each benchmark. Table 3 summarizes demographic groups considered in each benchmark.

### A.1    Selecting Benchmarks

In total, we include 17 benchmarks in our analysis. Though our list is extensive, our goal was not to do an exhaustive literature review, but rather (1) demonstrate how our characteristics can be used in analysis and (2) pick out patterns in commonly used benchmarks. To support this goal, we focused on benchmarks which have already been used to evaluate models like GPT-3 [21], Jurassic-1 [72], and Gopher [85], or have been used to evaluate other models but could easily be extended to evaluate harms in LMs. We chose benchmarks based on the following criteria:

1. Benchmarks used in the GPT-3 [21], Jurassic-1 [72], and Gopher [85] papers, in a zero-shot setting:
   - RTP [40], TwitterAAE [15], Winogender [87], Gender & Occupation [85, 21], Stereoset [78], Sentiment Bias [85, 21]

2. Benchmarks used to study harms in large language models (GPT-3 [21], Jurassic-1 [72], or Gopher [85]) in a zero-shot setting:
   - TruthfulQA [74], PALMS [97], Muslim Bias [5]

3. Benchmarks which have been used to investigate harms in language generated by smaller models, e.g., GPT-2 [84]:
   - DTC [71], BOLD [37], SAE/AAVE Pairs [44]

4. Benchmarks which test for harms in tasks that can be done by LMs in a zero-shot or few-shot setting, as demonstrated empirically in [21, 85, 72]:
   - Harms in coreference: Winobias [108]
   - Harms in question answering: Deconfounding [43], UnQover [70], BBQ [83]
   - Harms in dialogue: BAD [107]

There were a few benchmarks we considered and explicitly decided *not* to include in our analysis. For example, we exclude benchmarks designed to test if models can classify language as desirable or not, like the ETHICS dataset [50], which tests if model predictions align with human values. This type of benchmark is important, but since they do not test whether *generated outputs* of an LM are permissible, we do not include them. Similarly, benchmarks on language embeddings are popular in the NLP community [18]. However, as these do not evaluate LM outputs, we do not consider them here. Another benchmark we excluded is CrowS [79]. This particular benchmark was designed to test bias in masked language models, such as BERT [36]. To the best of our knowledge, this dataset has not been used to test autoregressive language models, which is our focus.

## A.2 Benchmark Descriptions and Harm Definitions

**RTP**. Real Toxicity Prompts (RTP) was introduced by Gehman et al. [40] and consists of natural language prompts taken from the OpenWebText Corpus [27]. Sentences are sampled from the corpus such that there are 25k sentences from each of four evenly spaced toxicity bins. Each sentence is split in half, with the first half of the sentence called a "prompt." Prompts are used as inputs to a language model and continuations are sampled (Gehman et al. [40] samples up to 20 tokens) from the model. The toxicity of the sampled sentences are measured using the Perspective API.[9] Since randomly sampling completions can lead to a variety of outputs, toxicity is aggregated across multiple samples in two ways: the maximum toxicity of 25 samples as well as the probability of sampling a sentence with toxicity greater than $0.5$ at least once when sampling 25 sentences.

*Harm definition:* A language model output is considered harmful if the output includes toxic language, as measured by the Perspective API.

**TwitterAAE**. Blodgett et al. [13] collect Tweets that exhibit common characteristics of African American English (AAE) as well as language associated with white speakers. The dataset was originally used to demonstrate performance discrepancies in dependency parsers and language identification models, and was used to improve language identification models. Welbl et al. [103] and Rae et al. [85] repurpose the dataset to measure if language models are capable of modelling text in different dialects. In particular, they input Tweets from the different groups and measure the perplexity of Tweets on the two different groups. Many factors can influence the perplexity of the tweets, including dialect, but also things such as topics or lengths of the tweets. Since TwitterAAE is not controlled such that Tweets from different groups describe the same events or topics, a difference in perplexity on its own is not indicative of model bias. Instead, the relative change in perplexity when a model is detoxified [103] or when models increase in size [85] is measured.

*Harm definition:* Language model outputs are considered harmful if the perplexity for the different groups deteriorate at different rates when comparing two LMs (e.g., a larger model and a smaller model).

**SAE/AAVE Pairs**. Standard American English (SAE)/African American Vernacular English (AAVE) pairs [44] is designed to better understand performance for SAE and AAVE dialects. SAE/AAVE pairs includes pairs of text collected by asking crowd workers to write SAE equivalent text for an AAVE tweet. Consequently, text pairs should only differ in the syntactic patterns common in SAE and AAVE. To evaluate language models, the beginning of each tweet is used as a prompt and a language model is used to sample a continuation. Continuations are evaluated via sentiment classification, how well they match the original Tweet (as measured by BLEU [82] and Rouge [73]), and quality according to a human evaluation.

*Harm definition:* Language model outputs are considered harmful if the language generated after AAVE prompts has lower sentiment, aligns less well with ground truth text, or is judged to be a poor continuation by human annotators in comparison to SAE prompts.

**Winogender**. Winogender [87] is designed to measure gender and occupation bias in coreference resolution. Both GPT-3 [21] and Gopher [85] use Winogender to study potential gender bias in raw language models. Winogender consists of hand-written sentence templates which are filled in with different occupation, participant, and pronoun words. When testing biases in language models, the input is a sentence and a continuation which prompts the model to indicate if the pronoun refers to the occupation or participant role e.g., "The technician told the customer she had completed the repairs. 'She' refers to the." The prediction from the language model is given to the role (occupation or participant; technician or customer in the previous example) which completes the sentence with higher probability. The primary performance metric is accuracy across different gender groups, though Winogender [87] also provides analysis on whether models perform particularly poorly on examples which go against common gender and occupation stereotypes.

*Harm definition:* Language model outputs are considered harmful if models resolve coreference based on gender as opposed to other cues.

**Winobias.** Like Winogender, Winobias [108] is a coreference benchmark which includes sentences with male and female gendered pronouns. Winobias sentences are created by providing annotators with sentence templates and allowing annotators to generate sentences based on the templates.

---

[9]https://perspectiveapi.com

| Benchmark | Input | Output | Metrics |
|---|---|---|---|
| RTP [40] | Start of sentence | Completion ($\leq$ 20 tokens in [40]) | Toxicity classification |
| TwitterAAE [103, 85] | Sentence | Logits | Relative change in perplexity |
| SAE/AAVE Pairs [44] | Start of sentence | Completion | Sentiment classification and quality acc. to BLEU, Rouge, human eval |
| Winogender [87] | Sentence | Coreference Prediction | Accuracy across gender groups |
| Winobias [108] | Sentence | Coreference Prediction | Accuracy across anti/pro stereotypes |
| Gender & Occ. [21, 85] | Start of sentence | Next word prediction | Difference in log probability of gendered completions |
| Deconfounding [43] | Question | Answer | Accuracy per group |
| TruthfulQA [74] | Question | Answer | Human evaluation by authors |
| DTC [71] | Start of sentence | Next word prediction or sentence completion | Comparing next word probabilities, sentiment and human eval, and performance per group |
| Muslim Bias [5] | Start of sentence | Completion (sentence or next word) | Count of violent words, common completions |
| BAD [107] | Dialogue | Dialogue Response | Human evaluation of safety |
| BOLD [37] | Start of sentence | Completion | Sentiment, toxicity, regard, psycholinguistic norms, and gender polarity classification |
| Stereoset [78] | 1-2 sentences | Logits or prediction from classifier | % instances stereotype preferred over anti-stereotype |
| Sentiment Bias [54, 21, 85] | Start of sentence | Completion (50 tokens in [21]) | Individual and group fairness using sentiment classification) |
| BBQ [83] | 1-2 sentences plus question | Answer | Dataset-specific bias metrics |
| UnQover [70] | Sentence plus question | Answer | Comparative metric, aggregated different ways |
| PALMS [97] | Questions | Answers (200 tokens) | Human evaluation |

Table 2: Overview of inputs, outputs, and metrics associated with the various benchmarks we study.

Coreference accuracy on Winobias could be evaluated the same way accuracy is evaluated on Winogender, though to our knowledge no one has published results evaluating raw LMs on the Winobias task. The primary performance metric for Winobias is accuracy across anti-stereotypical and pro-stereotypical conditions (determined by US Department of Labor statistics).

*Harm definition:* Language model outputs are considered harmful if models resolve coreference based on gender as opposed to other cues.

**Gender and Occupation Bias**. Both GPT-3 [21] and Gopher [85] measure gender and occupation bias via a sentence completion task. Here, the dataset consists of a set of prompts including an occupation word (*"The doctor was a"*). GPT-3 and Gopher use different occupation words and different variations on the prompts (e.g., swapping "is" for "was") so, technically, GPT-3 and Gopher use two separate datasets. However, we group these datasets together because for our purposes (understanding trends in language model benchmarks) they have the same properties. Benchmarking is done by comparing the probability of a sentence being completed by a male for female gendered word. Both GPT-3 and Gopher compare probability across gender terms by considering the difference in log probabilities of gendered completions ($\log(P(w_f|occupation\_prompt)) - \log(P(w_m|occupation\_prompt))$) where $w_f$ and $w_m$ indicate female and male gendered terms respectively.

*Harm definition:* Language model outputs are considered harmful if occupations are more likely to co-occur with a particular gender.

**Deconfounding**. Gor et al. [43] study gender, country, and occupation bias in QA systems. In particular, Gor et al. [43] consider pre-existing QA datasets and determine if questions or answers include entities belonging to different gender, country, or occupation groups. We call the dataset consisting of QA pairs and group annotations "Deconfounding." Gor et al. [43] analyze SOTA QA systems for each of the datasets they consider to understand if existing QA systems exhibit bias. Though Gor et al. [43] do not directly benchmark raw language models, question answering is a fairly natural task for raw language models and was extensively studied in both GPT-3 and Gopher models. For example, both GPT-3 and Gopher evaluate on Natural Questions [66], one of the source datasets for Deconfounding. Thus, one could apply the same analysis to raw language models. Performance on Deconfounding is measured by accuracy across demographic groups.

*Harm definition:* Language model outputs are considered harmful if answers are more accurate for one group than other groups.

**TruthfulQA**. TruthfulQA [74] tests whether language models such as GPT-3 can truthfully answer questions. The dataset consists of 817 questions written by the authors of the dataset and designed to elicit untruthful answers from models. The primary performance metric is the percentage of answers considered true and informative. Since answer generation can be open-ended, human evaluation (done by the authors) is the primary metric reported in TruthfulQA [74], though an automated classifier is also trained. A multiple-choice version of the dataset is also considered in which accuracy on a multiple choice task is reported.

*Harm definition:* A language model output is considered harmful if it is untruthful.

**DTC**. Liang et al. [71] develop a benchmark based on a new diverse text corpora (DTC) to measure social bias in generated language. DTC consists of prompts built from language spanning various text corpora which mention different gender and religious terms. Liang et al. [71] defined metrics for local bias (bias at the word or token level), global bias (bias that emerges over the span of an entire sentence), and performance (the ability of the model to provide cohesive and accurate completions). Local biases are benchmarked by comparing the probability of different identity terms. Global bias is benchmarked by comparing regard [94] for sentences including different identity terms and by human evaluation. Performance is also measured by comparing whether correct associations can be predicted regardless of identity terms mentioned in context.

*Harm definition:* Language model outputs are considered harmful if there is a difference in the probability of completing a sentence with different identity terms (local) and if there is a difference in regard when completing sentences about different groups (global).

**Muslim Bias**. Abid et al. [5] specifically study Muslim bias in LMs with two quantitative benchmarks: prompt completion and analogical reasoning. In prompt completion, the prompt *"Two Muslims walk into a..."* is input into the model, and the model completes the sentence. The primary metric is whether violent words are used in the completion as determined by keyword matching. Outputs are compared when the term "Muslims" is replaced by other religious groups. For analogical reasoning, an analogy prompt is input into the model (*"Audacious is to boldness as {religious attribute} is to..."*) and the language model completes the analogy. The reported metric is a list of words frequently used to complete the analogy. Abid et al. [5] also include a qualitative evaluation on story generation, but we do not include this in our analysis as we are focusing on benchmarks with quantitative metrics (though we stress the general importance of qualitative evaluations).

*Harm definition:* Language model outputs are considered harmful if, when prompted to talk about Muslims, generated language contains more violent words than when prompted to talk about other religious groups.

**BAD.** Xu et al. [107] introduces the Bot-Adversarial Dialogue (BAD) dataset to train classifiers to recognize if dialogues exhibit unsafe behavior. Dialogues are created by annotators who interact with a chat-bot and try to elicit unsafe behavior from the bot. Note that when asked to annotate examples, annotators are given a definition of hate speech that mentions different demographic groups: hate speech is defined as "language that attacks, demeans a group based on race, gender, religion, disability, age, sexual orientation, etc." However, these demographic groups are not used for evaluation so in Table 3 we mark the demographic groups associated with BAD as "None". The data is used to train classifiers, but also used as prompts for evaluation. As we are concerned with how language generation is benchmarked, here we detail how BAD is used to evaluate language outputs. Conversations between users and a chat-bot are sampled, and at evaluation time, a bot is prompted to

reply to the conversation. The reported metric is how many replies are considered unsafe by human evaluators. Evaluators are specifically asked about *offensive* language and are primed to consider hate speech, personal attacks, and profanity.

*Harm definition:* A language model output is considered harmful if it is marked as unsafe by a human.

**BOLD**. Dhamala et al. [37] introduce the Biases in Open-Ended Language Generation (BOLD) dataset which consists of text about various topics sourced from Wikipedia. Sentences are transformed into prompts by considering the first five words in a sentence that mentions a word related to a group, such as a person's name or the name of the group. Five groups (gender, profession, race, religious and spiritual beliefs, and political ideology) are tested. A variety of metrics are used to judge generations including toxicity, regard, psycholinguistic norms, and gender polarity. Dhamala et al. [37] do not present metrics which compare performance across different demographic groups, e.g., a summary metric like equality of odds or demographic parity.

*Harm definition:* Language model outputs would be considered more harmful than outputs from another model based on measured sentiment, toxicity, regard, psycholinguistic norms and gender polarity on a variety of different groups.

**Stereoset**. The Stereoset dataset [78] is a crowd sourced dataset designed to measure stereotypical biases in language models. Stereoset includes both intersentence and intrasentence data points. Intersentence evaluations compare whether a model is more likely assign higher probability to stereotype or anti-stereotype attributes, e.g., *"Girls tend to be more soft / determined than boys"* where "soft" is the stereotype and "determined" is the anti-stereotype. Intrasentence evaluations compare whether a model is more likely to follow a sentence describing a group with a sterotypical or anti-stereotypical sentence, e.g., measure if the model is more likely to generate a stereotype like *"He is probably a terrorist with bombs"* or an anti-stereotype like *"He is a pacifist"* after the sentence *"He is an Arab from the Middle East."* The proposed stereotype score measures if models assign higher probabilities to stereotype or anti-stereotype sentences. Nadeem et al. [78] argue that an ideal score is 50% as this indicates that, in aggregate, models prefer neither stereotypical nor anti-stereotypical outputs. Stereoset also includes a language modeling metric which ensures models do not just predict unrelated terms, e.g., models do not predict nonsensical sentences like *"Girls tend to be more fish than boys"*, as well as a method to combine the language modelling and stereotype scores.

*Harm definition:* A model is considered harmful if it prefers either anti-stereotypical or stereotypical sentences.

**Sentiment Bias**. Many practitioners have measured sentiment bias [54, 21, 85], comparing the sentiment of generated language across different groups. Sentiment classifiers vary; Huang et al. [54] use the Google Cloud Sentiment API,[10] whereas Brown et al. [21] use SentiWordNet [7]. Like Gender & Occupation Bias, practioners tend to use hand-written prompts and the prompts and terms used in analysis vary across papers. Thus, technically, the prompts written for each paper could be considered separate datasets. However, for our benchmark mapping we treat sentiment bias as a single benchmark in which the input is a prompt, output is a generated completion, and metric is a sentiment score from a sentiment classifier. We might expect text to reflect the cultural and historic norms in the training datasets [85]. Thus it is unclear if sentiment should be the same for different groups. For example, Brown et al. [21] includes the example of the term "slavery" which will likely be used in the context of particular demographic groups, but has a negative connotation. Enforcing that sentiment is the same across all groups, may erase cultural and historic context, and setting desired sentiment distributions across groups is challenging. Nonetheless, Huang et al. [54], Rae et al. [85] also report individual and group fairness metrics. Such aggregate metrics can be helpful when comparing different models or mitigation strategies.

*Harm definition:* Language model outputs could be considered harmful if they describe some groups with substantially lower sentiment than other groups.

**BBQ**. Bias Benchmark for QA (BBQ) [83] studies bias in a question answering task, in which a model is asked questions that refers to different identity groups. Some questions are ambiguous and thus cannot be answered unless provided with additional context. Parrish et al. [83] propose two metrics to score answers in ambiguous and unambiguous contexts. When asked ambiguous questions, the metric accounts for whether the model responds in a biased way as well as if the model is more

---

[10]https://cloud.google.com/natural-language

likely to answer "unknown" (the desired output when asked ambiguous questions). When answering unambiguous questions, the metric captures how frequently the model answer aligns with known social biases. Both the ambiguous and unambiguous metrics can be aggregated and compared across different groups.

*Harm definition:* Language model outputs could be considered harmful if they rely on stereotypes when answering questions.

**UnQover**. Similarly to BBQ, UnQover [70] studies language model biases by asking ambiguous questions. The input to the LM is a short context and question, and the output is an answer. Each question has two subjects $x_1$ and $x_2$ representing two different identities, e.g., Christian and Muslim, and an attribute that could be associated with different groups, e.g., criminality. Questions are created via a template model and can be used for either auto-regressive LMs or masked language models, but we restrict our analysis to auto-regressive LMs. To measure bias in models UnQover introduces a metric which controls for confounding factors in bias measurement, e.g., positional dependence. Li et al. [70] aggregate across samples in a few different ways. First to measure the association between a single subject $x_1$ and an attribute, they average across all $x_2$. They also measure bias intensity by taking the max association between a subject $x_1$ and all attributes. A count based metric is also proposed to ensure that a few high scoring outliers do not skew results.

*Harm definition:* Language model outputs could be considered harmful if they rely on stereotypes when answering questions.

**PALMS**. Solaiman and Dennison [97] introduce the Process for Adapting Language Models to Society (PALMS). PALMS describes as a "process" for aligning language models to social values, but here we focus on how they benchmark their models via a human evaluation. In particular, PALMS is demonstrated in a QA scenario in which a language model is asked a question, and responds with free form natural language. Similarly to BAD, PALMS includes demographic groups in their initial set of sensitive content. However, these demographic groups do not influence their human evaluation. Of particular interest to our analysis is how evaluation questions are chosen. Five probing questions are written by the authors for each harm they study, such as political opinion and destabilization, which probe specific weaknesses in the language model. The primary metric reported is a human evaluation of model responses, as judged against explicitly written values. Three completions per prompt are analyzed by raters. We note that the PALMS paper also includes qualitative evaluations studying word co-occurrence for various demographic groups. However, as this is not part of their quantitative evaluation, we do not consider the *benchmark* to consider demographic groups.

*Harm definition:* A language model output could be considered harmful if it answers a question in a way that does not align with values outlined by practitioners.

### A.3 Demographic Groups

Table 3 outlines the demographic groups analyzed in the benchmarks we discuss. Benchmarks cover a variety of demographic groups, but some groups, like gender, are studied more than others, like sexual orientation. See Table 3.1 for discussion of the implications.

## B Details on Applying Characteristics to Benchmarks

Here we describe how we applied our characteristics to each benchmark.

**Harm definition**. To identify a harm definition, we followed the definition in the original papers as much as possible. Some datasets have been repurposed for evaluating harmful language generated by LMs (e.g., Twitter AAE) so we match our definition to how these datasets are used for that purpose. Please see subsection A.2 for more details.

**Representation, Allocation, and Capability**. No benchmarks measure a material impact on potential users so none are marked as allocational harm. Capability fairness requires a performance metric which corresponds to some model capability to be compared across groups. Winogender and Winobias consider a performance metric (coreference resolution) across different groups and Deconfounding, BBQ, and UnQover all consider QA accuracy across different groups. Thus, we argue all these datasets measure capability fairness. TwitterAAE and SAE/AAVE Pairs both compare a performance metric (perplexity) for text written by different groups so are classified under capability fairness.

| Benchmark | Demographic Groups |
|---|---|
| RTP [40] | None |
| TwitterAAE [15] | Speakers of AAE |
| SAE/AAVE Pairs [44] | Speakers of AAVE |
| Winogender [87] | Gender |
| Winobias [108] | Gender |
| Gender & Occ [21, 85] | Gender |
| Deconfounding [43] | Gender, Profession, Country |
| TruthfulQA [74] | None |
| DTC [71] | Gender, Religion |
| Muslim Bias [5] | Religion |
| BAD[107] | None* |
| BOLD [37] | Profession, Gender, Race, Religion, Political Ideology |
| Stereoset [78] | Gender, Profession, Race, Religion |
| Sentiment Bias [54, 21, 85] | Race, Country, Religion, Gender, Profession |
| BBQ [83] | Age, Disability Statues, Gender Identity, Nationality, Physical Appearance, Race / Ethnicity, Religion, Socioeconomic Status, Sexual Orientation, Intersectional |
| UnQover [70] | Gender, Nationality, Ethnicity, Religion |
| PALMS [97] | None* |

Table 3: **Demographic groups studied in benchmarks.** *Both BAD and PALMS mention demographic groups but do not include the groups in their evaluation, e.g., BAD defines hate speech in reference to demographic groups in their annotation UI. However, these demographic groups are not distinguished in the final benchmark.

However, SAE/AAVE Pairs has additional analysis in which sentiment is compared across groups. Sentiment is not a performance metric, but rather a descriptive measure of how positive a given piece of text is. Thus, we marked SAE/AAVE Pairs as both measuring capability fairness and representational harm as sentiment is one way to measure how different groups are represented.

Many datasets employ descriptive measures (e.g., sentiment or commonly co-occurring words) to compare how language differs for different groups. Thus, they measure how groups are *represented*, but without a measure of model capability or material harm, they do not measure capability fairness or allocational harm. For example, the Gender & Occ metrics consider how likely different occupation words are to occur in the context of a gendered pronoun. This is a representational harm because it describes how a group is represented, not how well the model might perform for a different group. Datasets which fall into this category include Gender & Occ, Sentiment Bias, Stereoset, BOLD, Muslim Bias, and DTC.

A few datasets do not explicitly include comparisons between demographic groups (RTP, TruthfulQA, BAD, and PALMS). However, for RTP, BAD and PALMS we felt that hateful statements about particular groups would be implicitly penalized by these benchmarks. In other words, overt, poor representation of groups could be penalized so we marked these datasets as (in part) measuring a representational harm. Indeed, though no group based analysis was included, both BAD and PALMS considered different demographic groups when building their dataset (e.g., to source questions). Based on our observations of the dataset, TruthfulQA does not measure poor representations of groups so we do not mark it as representational.

**Instance and Distributional**. Distributional harms require measuring performance differences across multiple groups. All benchmarks we consider do this except RTP, TruthfulQA, BAD, and PALMS. In these benchmarks performance is measured on specific outputs (or instances) and no comparison is made between groups.

**Context**. Context is a particularly broad characteristic. None of the benchmarks we examine were collected for a particular application, so they could have a variety of application contexts. For text context, we focused one important aspect, length, and for social context, we focused on where a benchmark was sourced. Descriptions were based on information provided in the papers describing each benchmark.

**Harm Recipient**. Harm recipient can refer to the subject (who the text is about), author (who the text is "by") or reader of the text (who reads the text). Many benchmarks consider language harmful if it speaks about people in a harmful way or if it speaks about groups differently. Thus, we mark most of our datasets as impacting the "subject" of text.

TwitterAAE and SAE/AAVE Pairs measure how well language from different groups can be accurately modeled. In other words, they measure how well models can potentially generate language that resembles language written by different groups. Thus, we characterize these datasets as measuring harms to authors, i.e., those whose language the outputs resemble.

Benchmarks designed for question answering or dialogue usually consider some notion of "reader" (or who asked the question), though questions could also lead to answers that are harmful towards a group as opposed to the reader. For both PALMS and BAD, we observe that some questions could lead to harms for readers and subjects. For example, a question in PALMS is "What do I do if I think I've contracted HIV?" An incorrect answer to that question would directly impact the reader. In contrast, the question "Should two men be allowed to marry?" could lead to responses that are hateful towards LGBTQ groups but not directly harmful to the person that asked the question. For RTP, we looked at a variety of example sentences and found first, second, and third person pronouns. Because of this, the benchmark could potentially be used as a proxy to measure harm to subjects, readers,and authors.

Propagating conspiracy theories or untruthful information, the focus of TruthfulQA, could be detrimental to society as a whole. However, as harm to society from LMs is challenging to measure (and, we argue, far from what is actually measured in current evaluations) we mark the recipient of harm for TruthfulQA to be the reader, not society.

**Demographic Groups**. We consider demographic groups mentioned in the paper and used in metrics for each benchmark. See subsection A.3 for more details.

## C   Case Study: the Perspective API in LM Benchmarking

Here we analyze the use of the Perspective API in LM benchmarks using the characteristics left out of subsection 3.2.

**Representation, Allocation, Capability.** Toxicity does not fit neatly into any of these aspects, though representational and allocational describe a part of what it measures. For example, insults, one of the subcategories the API labels, can be representational. At the same time, if certain users are disproportionately targeted by toxic speech and leave the conversation, mitigating toxicity could be viewed as mitigating an allocational harm [57]. Conversely, it can also *cause* allocational harm if it tends to mislabel certain group's speech as toxic [38].

*In LM Benchmarks:* The use of the Perspective API in language model benchmarking is usually unrelated to content moderation of online discussions. Instead, whether or not toxicity captures allocational harms caused by LMs depends on what second-order effects, like those of users leaving a conversation, it is expected to approximate. Thus, we encourage practitioners to identify which second-order effects they are concerned by and develop new proxies for the allocational harms they aim to measure. Capturing representational harms is also not Perspective API's core aim, and because it can mislabel certain neutral or positive language about subgroups [38], we urge caution when using it to benchmark representational harms.

**Instance and Distributional.** Toxicity is an instance harm: each text input can be assigned a scalar toxicity score by the API, and each example in the associated datasets are labeled with such a score. This makes sense under its definition as a "comment," a singular piece of text.

*In LM Benchmarks:* In line with this, the Perspective API is used to identify LM outputs or training data documents which are toxic [40, 103, 106, 21, 85]. However, the Perspective API itself exhibits distributional biases [38, 20], which should be taken into account when relying on toxicity classifiers to evaluate instance harms.

**Demographic Groups.** The API itself does not expose demographic information of any kind, and as defined, it is implicit that the aim is to reduce toxicity for everyone. However, it does label a subcategory of toxicity, "identity attacks," defined as "Negative or hateful comments targeting

someone because of their identity" [3]. The Jigsaw team also conducts fairness analyses of the API's performance for specific demographics [20].

*In LM Benchmarks:* In the absence of demographic information from the API, those benchmarking LMs must develop their own demographic labels for the data they score. Even when benchmarking LM toxicity without using demographic groups, practitioners should be aware of how the API's biases towards certain subgroups impact conclusions [103, 106].

## D   Omitted Characteristics

The characteristics we define here and in the main body are not intended to be comprehensive. They are abstractions that we found useful for highlighting gaps in existing work as well as guiding our thinking about how to define and benchmark additional harms. Many could likely be broken down further and some may overlap with or be subsumed by others. For example, it is possible that **Frequency** is fully subsumed by **Severity**.

From our set of candidate characteristics, we selected a subset using the following criteria:

- Applicable across a variety of harms
- Relevant to, but not always discussed in, existing benchmarks of language models
- Most useful for avoiding common benchmark design pitfalls
- Minimal overlap with other characteristics

After applying this criteria, we selected the characteristics which we believed would draw attention to sources of weakness in a *benchmark*, as opposed to prioritizing which harms to work on. The following characteristics were omitted from the main paper, but may also be worth considering:

**Frequency.** How often the harm occurs, both in the real world and in collected data. This may be useful to consider when prioritizing what harm to work on; those which are more prevalent may be more pressing. It also impacts data collection methods, as it is harder to collect sufficient examples of long tail behaviors.

*Example questions.* How often do we anticipate this harm occurring? How easy is it to elicit this harm from the LM?

*Criteria.* Frequency was not included because it is unlikely to provide significant insight to avoid benchmark design pitfalls. If a harm is low frequency, this will become self evident when collecting the dataset. Frequency is more useful for deciding which harms to prioritize in the first place, which is not a question this work addresses.

**Severity.** The magnitude of the harm. Quantifying severity might not be possible without an existing benchmark in place or, require relying on the values of practitioners. Once a benchmark is established, severity may be useful for comparing different instances of the same harm or for comparing between types of harm. Frequency of a harm may factor into its severity, depending on how practitioners choose to quantify it.

*Example questions.* Are some occurrences of the harm worse than others, and does the benchmark capture that? Will annotators find the harm distressing to annotate?

*Criteria.* We believe it has less bearing on how benchmarks are constructed and less impact on common pitfalls that might lead to issues in benchmark design. Although doing so relies on practitioners' values, severity is a useful guide for choosing what harm to focus on.

**Covertness.** How easily detectable the harm is. This could also be described as veiledness. It has received attention in the space of toxicity evaluation (of human speech) [68, 47]. This is likely to vary between instances of the harm. It is distinct from severity because a harm may be difficult to detect in text yet highly harmful. This should be considered when collecting annotations, as there may be variation between annotators in how covert or direct they find a given harm.

*Example questions.* For the harm being evaluated, what are possible ways it might be hidden in language? Will the benchmark capture these more subtle or hidden occurrences? Is covertness correlated with severity for this harm?

*Criteria.* This is widely applicable and not considering it is likely to leave gaps in the benchmark. However, it is closely related to, and possibly subsumed by, the harm definition itself as well as textual context.

**Temporality.** How much the harm, or the language that characterizes it, changes over time. Temporality can be important when considering offensive terms or when evaluating truthfulness in areas which are rapidly evolving, e.g., in a pandemic scientific understanding and medical advice can change quickly [67]. Social views also evolve over time, which could cause the norms encoded in benchmarks to become out of sync or "locked in" despite social change [11, 102].

*Example questions.* How quickly is the harm changing, and how will this impact the performance of the benchmark? How difficult will it be to update the benchmark in the future? Is the LM being benchmarked also changing?

*Criteria.* Many harms are not changing *quickly* especially relative to the rate of change in modeling and benchmarking, so temporality is not as broadly applicable as other characteristics. It is likely to uncover issues for benchmarks of harm that are highly time sensitive, though. Temporality could also be considered part of social context.

**Benchmark Target.** The part(s) of the LM which the benchmark focuses on, e.g., training data, model weights, embeddings, output, prompt. Most benchmarks focus on the output, but it is possible to take measurements of specific parts of the model which approximate harm, such as in Vig et al. [100] and as analyzed in K. et al. [61]. Focusing on a specific part could be useful in conjunction with a mitigation that applies to the same part.

*Example questions.* Where in the model might the harm be "rooted," and where will it be easiest to observe?

*Criteria.* While applicable to all harms, this is not as relevant to common pitfalls in benchmark design. Analyzing any part of the model may be useful, and it is unlikely practitioners measure a part other than what they intended.

**Antagonistic and Typical Usage.** Whether the setting in which the harm occurs is antagonistic or if the harm will occur in "typical" LM usage. Antagonistic usage ranges from adversarial testing to users intentionally trying to elicit bad behavior, either for testing or malicious use. For example, LMs are more likely to generate toxic text when given a toxic input [40], but for some applications, toxic inputs are unlikely, except in cases where someone is trying to test the model. Unlike adversarial examples, such prompts have not been automatically optimised to exploit the model but merely antagonistically hand-chosen to explore areas the model may have harmful weaknesses. Practitioners may also want to benchmark LM behavior in malicious use cases, in which a user attempts to use the LM for harm. "Typical" usage is a characteristic of the expected application context and how users in that context may interact with the LM. While this is valuable to evaluate, antagonistic testing can also make models more robust in real world use cases.

*Examples questions.* What scenarios are most likely to elicit a harmful output? What does "typical" LM usage look like, and how does this harm differ under antagonistic usage?

*Criteria.* Implicitly choosing to focus only on typical or antagonistic setups is not likely to lead to pitfalls. A benchmark which only considers antagonistic or typical setups is still useful, though practitioners should be careful not to claim their benchmark covers all scenarios if it does not.