# OpenReview forum: "Characteristics of Harmful Text: Towards Rigorous Benchmarking of Language Models"
_NeurIPS.cc/2022/Track/Datasets_and_Benchmarks — NeurIPS 2022 Datasets and Benchmarks _

### Official Review · Reviewer_8rZR · 2022-07-18
**interesting discussion but without solid results**

**Rating:** 5
**Confidence:** 3
**Clarity:** The paper is well-written and clearly…

**Strengths:**

The paper is well-motivated. As indicated in the paper: a single benchmark cannot cover all harms, but their characteristics allow explicit understanding of what a benchmark might (and might not) capture. Also, the characteristics enable the user to analyze whether these benchmarks measure what they claim to.

The authors provides 17 benchmarks in their analysis to demonstrate how their characteristics can be used in analysis and (2) pick out patterns in commonly used benchmarks.

**Weaknesses:**

The paper is interesting and provide many insight that can be valuable in benchmarking language model. However, there is no concrete metrics/measures towards rigorous benchmarking but only discussions on the six characteristics. Also, both the case study and the 17 selected benchmarks do not fully reflect the discussion of the characteristics earlier.

The manuscript should be self-contained, rather than asking the readers to go to Appendix ABC for more details. The main paper looks like an extended abstract of the appendix right now.

**Additional Feedback:**

It would be great for the authors to apply all of their six characteristics to the case study (and the 17 benchmarks) and discuss their role and their interactions in terms of the harmful text. The discussion in section 3 may be too generic.

The last column of table 2 is messy and hard to read.

**Correctness:**

The evaluation methods and experiment design are appropriate and performed correctly. However, there is no solid proof on whether and how much the proposed metric/measures can work and serve the benchmarking purpose.

**Documentation:**

NO. This reviewer believes there is insufficient detail to support reproducibility and the analysis on 17 benchmarks are dataset dependent and subjective.

**Ethics:**

The paper focus on characterizing harmful text in benchmarking language model. And this could help alleviate the misuse/harm of the language model in the design of language model, which is helpful from the perspective of the ethics.

**Relation To Prior Work:**

Yes. The authors clearly discussed how each of their characteristics related to/originated from previous contributions.

**Summary And Contributions:**

This work outlines six ways of characterizing harmful text which merit explicit consideration when designing new benchmarks:
harm definition; representation, allocation, capability; instance and distributional; Context: textual. application, social, harm recipient: subject, reader, author, society; demographic groups.
The authors then use these characteristics as a lens to identify trends and gaps in existing benchmarks and  conduct case study of the Perspective API. The paper bridge between foresight and effective evaluation.

---

> ### Author Response · Authors · 2022-08-24
> **Thank you, response and clarification**
>
> We were pleased to read that the reviewer believes “the paper is interesting and provide many insight that can be valuable in benchmarking language model.”  Below we address concerns from the reviewer:
>
> *However, there is no concrete metrics/measures towards rigorous benchmarking but only discussions on the six characteristics.*
>
> Developing frameworks and taxonomies is an important step in its own right and intended to be a foundation/ platform for further research to then develop individual benchmarks.  It's valuable to bring together these characteristics in a single horizontal, cross-cutting overview so the community can identify potential gaps in benchmarks.  In its short history, the NeurIPS Benchmarks proceedings have included work that laid out concepts, overviews, or taxonomies without providing (new) quantitative evidence including:
>
> AI and the Everything in the Whole Wide World Benchmark
> Teach Me to Explain: A Review of Datasets for Explainable Natural Language Processing
> Are We Learning Yet? A Meta-Review of Evaluation Failures Across Machine Learning
> Artsheets for Art Datasets
>
> *It would be great for the authors to apply all of their six characteristics to the case study*  We included all six characteristics for the toxicity case study in Appendix C in the original submission.  We omitted a few in the main paper due to space limitations.
>
> *This reviewer believes there is insufficient detail to support reproducibility and the analysis on 17 benchmarks are dataset dependent and subjective.*
>
> We have added details to the appendix (section B).

---

### Official Review · Reviewer_M3My · 2022-07-23
**Although the paper tries to benchmark models to characterise harmful text, I am not convinced by the rigour of the benchmarking pipeline.**

**Rating:** 4
**Confidence:** 3

**Strengths:**

These characteristics can help in making robust language models.


**Weaknesses:**

1) Though several characteristics have been presented in the paper, some of these are already known to the research communities, such as harm definition, representational and allocational harm, etc., based on prior research. So re-introducing them may not be much helpful for benchmarking.
2) How characterization of different benchmarks mapping has been done is not clear.
3) No quantified analysis has been done on any language models mentioned in the paper. It would have been better to show some quantified results of a few LMs to further make others understand the weakness of the existing benchmarking.
4) Even for the case study, how Perspective API has its own challenges discussed via fuzzy approaches based on the existing literature. Again quantified results are needed to claim the argument.

**Additional Feedback:**

N/A

**Clarity:**

Somewhat clear.


**Correctness:**

The evaluation of the existing benchmarking models/datasets is not established precisely. Quantitative analysis is required.


**Documentation:**

N/A

**Relation To Prior Work:**

N/A

**Summary And Contributions:**

The author introduced six characterizations of harmful text to benchmark language models in this paper. The paper further showed the characteristics that are missing from the existing benchmarks. These characteristics have been applied to Perspective API as a case study to demonstrate its associated challenges.

---

> ### Author Response · Authors · 2022-08-24
> **Thank you, response, and clarifications**
>
> Thank you for your review!  Below we address the concerns raised about our paper:
>
> We politely disagree that “re-introducing them [the characteristics] may not be much helpful for benchmarking.”  Though some of the characteristics we discuss are mentioned in prior literature, we believe our paper contributes to the current literature in the following ways:
>
> -- As discussed in L24-L29, the ML fairness community has documented a variety of sociotechnical insights, and we consider these within the context of language benchmarks specifically.  For example, much work in ML fairness has discussed demographic groups, but there are particular considerations (e.g., marking) which are important for building language benchmarks.
>
> -- Collating different characteristics for data collection to allow for holistic understanding of potential pitfalls.  Characteristics are not independent (L353 of the original submission and L364 of the updated submission) and different characteristics impact each other.  This is exemplified by some of our “example questions”.  In L97 in the original submission L104 in the updated submission (“Does the type…match the type implicit in the harm definition?”) illustrates how instance and distributional harms might impact harm definition and L84 in the original submission L87 in the updated submission  (“How is this harm…distributed across groups?”) points to how representational and allocational harms are related to demographic groups.
>
> -- Finally, using our analysis we are able to point to gaps in the field and illustrate how carefully considering characteristics can illuminate potential pitfalls in various benchmarks (Sections 3.1 and 3.2).
>
> *How characterization of different benchmarks mapping has been done is not clear.*  Thanks for the comment!  We added in a description to the appendix and believe this makes our submission stronger.
>
> *No quantified analysis has been done on any language models mentioned in the paper* and *Again quantified results are needed to claim the argument*
>
> We believe our paper is clearly within scope for NeurIPS Benchmarks papers: “In addition to new datasets and benchmarks on new or existing datasets… **Frameworks for responsible dataset development**, audits of existing datasets, **identifying significant problems with existing datasets** and their use, or systematic analyses of existing systems on novel datasets that yield important new insight are also in scope.”  We believe our work falls under both frameworks for responsible dataset development and identification of problems with existing datasets.  We draw extensively on quantitative results reported in the literature and do not believe adding more quantitative results would increase the validity of our work.

---

### Official Review · Reviewer_T9zc · 2022-07-26
**Unclear scope and contribution**

**Rating:** 3
**Confidence:** 3

**Strengths:**

The submission is neither a dataset or benchmark, and its strengths must be described on its own terms rather than the typical criteria of the track. The analysis framework it presents is exhaustive and sheds light on distinctions that might be overlooked. The limited coverage of some of the areas it describes is a useful waypoint for future work to cover. The vocabulary it introduces is clear, unifies existing work, and could be useful for future work.

**Weaknesses:**

The submission does not introduce new benchmarks or datasets. It identifies areas of measurement as under-covered but does not introduce new benchmarks or modifies existing ones to cover them. Its main novel contribution is casting existing work under the same framework and vocabulary. It is not backed by new experiments.

**Additional Feedback:**

I am not sure this work is in scope for the datasets and benchmarks track. It is not a dataset or benchmark, nor falls in any of the categories defined in the scope section of the call for papers. It reads like a vision paper, or a review paper, about harm benchmarks, which might be of interest to this track even if it is not explicitly in the scope.

**Clarity:**

The paper has clear structure and writing. The different possibilities for every aspect of harm are well described and easily comprehensible.

**Correctness:**

The submission does not make falsifiable claims. The analysis of the Perspective API seems sound, although it does not test its predictions (for example, the remark that it is used on text that distributionally differs from its training data could be experimentally backed)

**Documentation:**

This work does not present a dataset or benchmark, and does not have the same documentation needs. One example of application of the framework to an existing harm measurement is provided for reference, and it would be reasonably easy to perform the same analysis on any benchmark, although results might vary depending on who is doing the analysis.

**Ethics:**

I do not have ethical concerns with this work.

**Relation To Prior Work:**

Prior work is extensively cited. However, it is unclear how the work differs from previous contributions.

**Summary And Contributions:**

The submission draws upon existing literature on machine learning fairness to create a framework of analysis of harm benchmarks for language models. It applies this framework to a list of existing benchmarks and finds that their coverage is limited to a fraction of the identified possibilities for harm. It also analyses a public API for harm measurement in a more detailed way as a case study.

---

> ### Author Response · Authors · 2022-08-24
> **Thank you, response, and clarifications**
>
> Thank you for your kind review!  We are pleased that you found our framework extensive, clear, and potentially useful for further work.  We would like to address some of the concerns raised in the review:
>
> The main weakness cited is that our work “does not introduce new benchmarks or datasets”.  While this is true, we believe our work is in scope for the NeurIPS benchmarks and datasets track which explicitly states: “In addition to new datasets and benchmarks on new or existing datasets… **Frameworks for responsible dataset development**, audits of existing datasets, **identifying significant problems with existing datasets** and their use, or systematic analyses of existing systems on novel datasets that yield important new insight are also in scope.”  We believe our work falls under both frameworks for responsible dataset development and identification of problems with existing datasets.
>
> Another critique of our work is that we do not test our predictions: Though we do not provide new analysis on models, we draw extensively from current literature to make our claims.  For example, addressing the point about distribution mismatch for toxicity, we cite experiments showing that toxicity is higher for the books data in MassiveText than other slices (L298-302 of the original submission and L308-310 in the updated submission).
>
> Finally, we believe our work differs considerably from other work in the field and offers new insights.  Starting at L338 in the original submission and L349 in the updated submission we briefly outline how our work relates to other similar work in the field.  In particular, as stated in L339 in the original submission and L349 in the update submission, we believe that though extensive work has been done highlighting potential harms of LMs (e.g,. Bender, Gebru et al [10] and Weidinger et al. [99]), identifying a potential harm and defining it is only the first step in creating a harm benchmark.  Though other work has flagged some potential pitfalls (e.g., Blodgett et al [15]) our work offers a first attempt to unify a variety of pitfalls and reflective questions for those collecting new datasets aiming to measure language model harms.  As mentioned, our framework allows us to highlight current gaps in the field (L37), uncover why certain metrics might fail to capture what they intend to capture (L42), and offers one way those who create benchmarks can be explicit about what they measure.

---

### Official Review · Reviewer_Ag2r · 2022-07-26
**A position paper about benchmarking LM on potential (social) harms**

**Rating:** 7
**Confidence:** 3

**Strengths:**

Overall, the paper has merit, as it promotes a discussion about important issues in modern application of language models. I believe the paper should make more stress on that harm is often a subjective issue, and  should be treated with that perspective in mind. Some parts in the appendix are conductive to this, maybe make that a bit more prominent.

**Weaknesses:**

Transformers models are not discussed (though mentioned).
The paper restricts to autoregressive models, as those are often used in generation.
The choice is reasonable, but still a weakness.

**Additional Feedback:**

Some technical remarks about the manuscript:

I got the manuscript in two parts (appendices are in a separate file), and the hyperlinks from main manuscript to appendices file do not work. This should be solved if the manuscript goes to publication.

Also, the bibliographic reference list (105 entries) appears in the manuscript, and is repeated in the appendix file. It is probably not needed twice, once is enough.

Item 98 in the references list, a book by Linda Waugh, is missing publication data.

Table 1 – the headers are cryptic. What is "Rep.Cap.Alloc." ?  And "Dist.Inst."?  Please fix that.

Lines 264-265:
" we found that gender bias in LMs varies when between  gender terms like “female” vs. “girl.”
This sentence seems broken, "when" seems to be erroneous. .

Lines 298-299
"For example, the Books slice of the MassiveText dataset... "
That data set seems to be proprietary. The authors migh need to mention that explicitly.


Line 1219 (Appendices file) "...and as analyzed in K. et al. [58]. ..."  Bad reference ,fix it

Line 922 (and 926) "a white-aligned corpus."  This term seems weird and also a bit harmful. What exactly is a white-aligned corpus?   Try to use a more precise term, and also without bad connotations.

Line 926: "corpuses".  It is a correct form, but I think the community is more used to "corpora".


**Clarity:**

The paper is mostly clear, though it seems to presume familiarity with the topic of harm and the recent research.
The exposition could be made more reader-friendly, if it were intended to a general audience of NLP researchers.

**Correctness:**

As this paper is more like a position paper,  "Correctness" category does not seem to apply.
Some pieces in the paper can be more debatable  (especially what is harm, how to measure it),
but I see no problems of 'correctness'.

**Documentation:**

This is not applicable, as the paper itself is a 'survey' of various datasets, but it does not provide a new data set.

**Ethics:**

No problems here.


**Relation To Prior Work:**

Prior work is clearly discussed and referenced.

**Summary And Contributions:**

This is a very interesting paper.
It focuses on the issue of potential harms that can arise from the usage of modern large scale language models (LM). The paper specifically restrict their attention to autoregressive models, which a re typically used for language generation tasks.
 Although the authors say it is not a review paper, the paper (actually the appendix) does present a nice review of multiple published works that dealt with harm, and specific datasets or language models that were used in those papers. This by itself is already a valuable contribution, as it presents an up-to-date of relevant work. The authors were so nice to even reference works that were not included in the 'survey', so it is valuable collection of references.

The paper seems to be a kind of a mix between a mini review and a position paper. It discusses various aspects of harm that can arise with LMs, how such harms were defined and operationalized in various research works, and also how those harms can be 'classified' in some very broad terms.
Or, as the paper says "synthesizing existing  critiques of benchmarks and taxonomies of harm".
The paper further provides a discussion/examination of those aspect in relation to particular implemented system, Perspective API, that can be used for harm detection.

---

> ### Author Response · Authors · 2022-08-24
> **Thank you, response, and clarifications**
>
> Thank you!  We are pleased that you found our paper interesting and indicate it will be a valuable contribution.  We updated our paper to reflect minor clarity issues as mentioned in “additional feedback”.
>
> *Transformers models are not discussed (though mentioned). The paper restricts to autoregressive models, as those are often used in generation.*
>
> We did not describe transformer models and the differences between autoregressive and BERT models due to space constraints.  If the reviewer finds it helpful, we can add a section in the appendix for a final version.
>
> Re “white-aligned corpus”: we were following prior work that has used similar termonology (e.g., "Demographic Dialectal Variation in Social Media: A Case Study of African-American English" describes the language as “white-aligned language”).  We most definitely did not mean to cause offense and removed any usage of “white-aligned corpus” in our paper.  In particular, we updated the Appendix to read: “...African American English (AAE) as well as language primarily representing white speakers.”

---

### Official Review · Reviewer_k8Xw · 2022-07-27
**Characteristics of Harmful Text**

**Rating:** 7
**Confidence:** 3

**Strengths:**

The proposed characteristics are important and should be considered by future studies in abusive language detection, which is often used to control language generation. The included ethical reflections are important and may assist with social implications that are currently not considered enough.

**Weaknesses:**

Two weaknesses exist in this study, which, however, can be addressed.

1. This study is limited to the English language not only due to the difficulty of transferring cultural norms and translating benchmark assumptions but also due to the poor toxicity classification performance and under-representation in annotation platforms for other languages. The latter two could be addressed. Hence, the importance to do so should be highlighted. Otherwise, such limitations may become the norm and may motivate even less focus on other languages.

2. To demonstrate the use case of the proposed characteristics a toxic language classification model is assessed, not a language model. This is motivated by the fact that Perspective is often used as an LM building block. It is true that a toxicity classification component (not necessarily Perspective; fine-tuning BERT, for example, is another popular choice) complements LMs, but then the focus (incl. the title) of this study could be on toxicity classifiers, not on LMs.

**Additional Feedback:**

**Questions**
1. How do the characteristics of harmful text depend on the classification accuracy and the annotation capacity?
2. The short context (Table 1) has already been studied (e.g., the lack of conversational context for toxicity classification was already discussed in ACL 2020). Pointing out this weakness is in the right direction, but shouldn't we move to suggestions regarding how to bypass this known problem?
3. The authors conclude that the proposed dimensions should be used to change the operationalised definition from "toxic" to "toxic to who", and I agree. However, the emphasis of this study is on the (sociotechnical) language and it is not clear to me why is it limited to LMs. For example, this study could as well be focused on human-generated harmful language, which may be important to consider given that human annotators create machine-encoded commonsense knowledge. Can the authors please elaborate?

**Comments**
* Line 1: "that" points to text, hence: drive > drives
* Lines 32-34: Capitalisation seems strange to me
* Line 54: order > order to
* Lines 106-108: Just noting here that this may pose a significant challenge to systems, considering that context-dependent posts will be a minority of the -already minor- harmful class (discussed in ACL 2020)
* Line 115: include > includes
* Line 2814: text > a given text
* Line 300: the > that the
* Line 322: repetition of "conclusions" seems strange to me.


**Clarity:**

This paper is very well written. I would appreciate clarification on the following two points (minor comments shared as additional feedback):
* Line 120-122: Not clear to me how could connectivity issues lead to data exclusion in a not far-fetched scenario. Please consider elaborating.
* Line 310: Please clarify (if true or rephrase if not) that "the author of the comment that turned another reader leave the discussion"


**Correctness:**

The use case employs a single classification API. Not including any LM in such a study doesn't sound right. On the other hand, assuming that classifiers and not LM are in question (see weaknesses), at least another classification model could be employed to allow more informative conclusions to be drawn.

**Documentation:**

LMs are not benchmarked (Line 274) while Perspective API (often a component of LMs) is more analysed based on the proposed characteristics rather than benchmarked.

**Ethics:**

This study is limited to the English language, due to the difficulty of transferring cultural norms and translating benchmark assumptions, but also due to the poor toxicity classification performance and under-representation in annotation platforms for other languages. The latter two, however, could and perhaps should be addressed. The importance to do so should be acknowledged or such limitations may become the norm and may motivate even less focus on other languages.

**Relation To Prior Work:**

The relation to prior work is clear.

**Summary And Contributions:**

This paper presents six characteristics of harmful (English) text to be considered when assessing (English) language generation models. The authors focus on large language models, but this study could be related to any language generation benchmark, at least in principle. The proposed characteristics are used to identify where exactly existing models lack. Based on Table 1 of the study, these are the short textual context; the use of the subject (third person pronouns) as the recipient of the harm; and the limited number of benchmarks considering the potential harms due to a single output of the model (instance/pointwise harm), as opposed to distributional harms measured on evaluation sets of instances. Finally, a case study is performed on a toxic language classification API.

---

> ### Author Response · Authors · 2022-08-24
> **Thank you, response, and clarifications**
>
> We thank you for your positive rule and are pleased you believe our work “should be considered by future studies” and that “ethical reflections are important”.  Below we answer questions and critiques posed in your review.
>
> *“This study is limited to the English language”*.  We wholeheartedly agree.  We mention this in L237-L243 of the original submission and L246-252 of the updated submission but have updated this section to explicitly state how important benchmarks in non-English languages are.  In our abstract we now specify we are analyzing English benchmarks and an English toxicity classifier.  We also added a footnote in the introduction mentioning that our work primarily focuses on English benchmarks.
>
> *“To demonstrate the use case of the proposed characteristics a toxic language classification model is assessed, not a language model”* As mentioned in L274 of the original submission and L284 of the updated submission, we focus on the Perspective API as it is used to analyze many models.  In particular, the RealToxicityPrompts benchmark relies on the Perspective API and the Perspective API was used to study toxicity in Gopher and a similar toxicity classifier was used to analyze the Anthropic Assistant in (Askell et al. “A General Language Assistant as a Laboratory for Alignment”).  We expand on how decisions made when training the PerspectiveAPI impact LM harm benchmarking:
>
> -- **Harm definition**: Section E.3 of Challenges in Detoxifying Language Models point out specific ambiguities found when annotating LM model outputs for toxicity.  In particular, it is noted that annotation instructions were designed to make it “easy to compare our results against PerspectiveAPI scores”.  Thus, by being locked into the definition of the PerspectiveAPI, it restricted the way in which LM outputs could be evaluated.
>
> -- **Context**:  As mentioned in L298 of the original submission and L308 of the updated submission, it is unclear that the Perspective API is well aligned with the kind of data used to train LMs like Gopher.  Consequently, it is unclear what kind of conclusions can be drawn from comparing toxicity across models trained with this dataset.
>
> -- **Harm Recipient**:  Because the PerspectiveAPI is trained on content moderation data, it might enforce norms for discussions between humans that may not apply to LM text.  Though there are definitely overlaps in who can be harmed in content moderation and by LM generated text, they are different.  For example, as pointed out in L319 of the original submission and L329 of the updated submission, in content moderation, it can impact authors if their text is overly flagged as toxic and it is not clear how this carries over to LM generated text.  These differences should be acknowledged when evaluating language from LMs.

---

> > ### Author Response · Authors · 2022-08-24
> > **Thank you, response, and clarifications  (2)**
> >
> > Please see our answers to your other questions below:
> >
> > Clarity: *Not clear to me how could connectivity issues lead to data exclusion in a not far-fetched scenario*.  Much of the world has limited access to the internet (e.g., see https://www.internetsociety.org/issues/access/) and people from those communities might not write articles that would be scraped during dataset creation.  Hence, those languages/cultures may not be well represented in training datasets.  We felt “internet access” might be clearer than “internet connectivity” so updated this in our submission.
> >
> > *How do the characteristics of harmful text depend on the classification accuracy and the annotation capacity?*  Different characteristics might interact with classification accuracy and annotation capacity in different ways.  For example, if a benchmark is designed to capture a distributional harm, it might be more important that a classifier has similar accuracy across specific demographic groups than for the classifier to have higher overall accuracy.  For annotation capacity, characteristics that might be particularly important to consider are demographic groups and context.  If one group has more capacity than another, how will that be reflected in the final benchmark?  Likewise, if annotations are done in a particular context, we might need to reannotate to measure a harm in a different context.
> >
> > *The short context (Table 1) has already been studied (e.g., the lack of conversational context for toxicity classification was already discussed in ACL 2020). Pointing out this weakness is in the right direction, but shouldn't we move to suggestions regarding how to bypass this known problem?*  Thank you for the pointer – could you please specify the paper(s) from ACL 2020 you have in mind?  In Table 1, we also see that many benchmarks rely on handwritten prompts from practitioners and it is likely that these two issues are related, perhaps because generating and annotating harmful language requires careful thought not amenable to current annotation techniques.  One way forward is to consider how we incentivize annotators and invest in techniques that might yield higher quality data for long text annotations.  For example, more bespoke processes (e.g., the deliberative method detailed in Annotating Online Misogyny or the extensive quality control detailed in QuALITY: Question Answering with Long Input Texts, Yes!) might allow for more nuanced and careful annotations of longer documents.
> >
> > *…the emphasis of this study is on the (sociotechnical) language and it is not clear to me why is it limited to LMs…*  We chose to focus on LMs because we anticipate that the community will continue to build improved LMs and, at this point, it is unclear what the right evaluations are for these models.  Though some of our characteristics could be applied to human language, there are fundamental differences when thinking about machine and human generated language.  For example, as discussed in L311-321 of the original submission and L322-332 of the updated submission, LMs cannot be in-group and there is no protection for LM free speech.

---

### Official Review · Reviewer_6EL5 · 2022-07-28

**Rating:** 7
**Confidence:** 4

**Strengths:**

- The paper attempts to fill in an important gap in the study of harms and biases generated by (large) language models.

- This work has the potential to be of value and interest to the growing literature on the characteristics of texts generated by large language models. It would be nice if the NLP community would adopt the characterizations outlined in this paper moving forward.

- The paper’s case study nicely illustrates the mismatch between the Perspective API’s original design principles and its effective evaluation mechanism.


**Weaknesses:**

First of all, I believe the paper has some foundational issues: The authors fail to provide clear definitions of key terms that are used in the paper such as harm, bias, and toxic. I was initially hoping that the authors might use Section 2 to clarify the meanings of these terms, but these terms have been used throughout the paper without proper explanations or exemplifications, thereby leaving the readers to come up with their own definitions of these terms. Since the main focus of the paper is to characterize harmful text in natural-language processing and processing benchmarks, I believe it is vital that the authors clearly and concisely describe these terms. The authors might want to refer to “Data Statements for Natural Language Processing: Toward Mitigating System Bias and Enabling Better Science” by Bender and Friedman (TACL 2018) as an exemplifying paper: As they might see from this paper, Bender and Friedman first provide definitions of key terms used in the paper—such as  “NLP dataset,” “NLP system,” “dataset curator,” “technical bias,” etc.—before proposing their data statement schema. The authors might want to follow a similar strategy in their own paper, as the current version of the paper leaves the reader with a slight sense of confusion over the specification of these terms and might lead to some misunderstandings.

The authors seem to have spent a considerable amount of time outlining six ways of characterizing harmful text—and I, for one, would like to congratulate them on their tenacious work, but Section 2, as it stands, is still hard to follow, confusing, and incomplete. Let me provide elaborations on this point:

- Section 2.2: How does allocational harm differ from representational harm? Where do these harms (or biases) stem from?

- Section 2.2: The term “capability fairness” is truly confounding.
- Section 2.3: I really like the distinction between “instance” harm and “distributional” harm, but I believe this distinction was previously made by (Rae et al., 2022) [Scaling Language Models: Methods, Analysis & Insights from Training Gopher; Section 5.2 (pg. 14)], (Chowdhery et al., 2022) [PaLM: Scaling Language Modeling with Pathways; Section 10 (pg. 39)], and (Perez et al., 2022)[Red Teaming Language Models with Language Models; Section 6 (pg. 9)]. I would encourage the authors to look at these studies and cite them whenever needed.

- Section 2.4: I believe the contextual harms seem to contain a narrow and incomplete set of items. For instance, a humanist scholar (say, a sociologist) might find this categorization rather limited in its scope. What about cultural context? What about socioeconomic context? What about linguistic context? The authors seem to have skipped all these categories in their discussion.

- Section 2.5: Again, the harm recipients may not be limited to subject(s), author(s), and the society as a whole. Looking at (Bender and Friedman, 2018), the authors might realize that there might be other stakeholders who might be both the recipient and the creator of these harmful texts: The annotators, the curators, and the data owners, among some other groups.

- Section 2.6: Here the discussion can be made more specific and clear (again, as an example, please see Section 5 of (Bender and Friedman, 2018)).

While I appreciate the authors’ efforts to study the different characteristics of different NLP benchmarks such as Winobias, Muslim Bias, and PALMs using their proposed outline, and to include a case study to illustrate how the work can be used as a guiding book to examine a widely used classifier (Perspective AI) in Section 3, I believe this section could have been stronger if the authors were to justify or explain their characterizations in Table 1, say, in the Appendix. At the moment, the discussion in Section 3 seems too surface-level and superficial; I think the readers might appreciate it more if there were a more in-depth and critical analysis.

**References:**

Bender, Emily M., and Batya Friedman. "Data statements for natural language processing: Toward mitigating system bias and enabling better science." Transactions of the Association for Computational Linguistics 6 (2018): 587-604.

Rae, Jack W., et al. "Scaling language models: Methods, analysis & insights from training gopher." arXiv preprint arXiv:2112.11446 (2021).

Chowdhery, Aakanksha, et al. "Palm: Scaling language modeling with pathways." arXiv preprint arXiv:2204.02311 (2022).

Perez, Ethan, et al. "Red teaming language models with language models." arXiv preprint arXiv:2202.03286 (2022).


**Additional Feedback:**

**Additional suggestions and comments:**

- Line 15: customer service → customer-service

- Line 21: “current evaluation tools are imperfect.” → Can you elaborate on this point? In what ways are they “imperfect?”

- Line 22: “…is supported by our own work analyzing the Gopher model [82], in which we observed a variety of …” → “is supported by the work analyzing the Gopher model [82], in which the authors observed a variety of …”

- Line 54: in order concretize → in order to embody [alt]

- Line 68: Representation, Allocation, Capability → Representation, Allocation, and Capability

- Lines 71, 75, 80 (and some other lines): real world → real-world (since used as an adjective preceding a noun, need to use a hyphen)

- Lines 104-05: Some might also consider such a statement as noxious or insalubrious.

- Line 125: Harm Recipient: Subject, Reader, Author, Society → Harm Recipient: Subject, Reader, Author, and Society

- Line 134: group → a group

- Line 183: author → author(s)

**Clarity:**

While the submission is mostly clear, it could still be further improved if the authors could reorganize Section 2 and Section 3, describe the key terms and concepts used in the paper (such as “harm” and “bias”) at the beginning of Section 2, and provide more critical and in-depth discussions in Section 4.


**Correctness:**

Overall, the claims in the submission seem to be correct to me (though I must admit that the authors could have done a better job of corroborating some of their arguments in Sections 2 and 3).

**Documentation:**

N/A.

**Ethics:**

N/A.

**Relation To Prior Work:**

I am afraid that the authors have not discussed, at least in depth, how their present work differs from previous contributions (or survey papers on this topic). Neither have they discussed how their work is related to other prior studies that have looked at the inherent (distributional) biases encoded in large-scale language modeling corpora (such as OpenWebText and the Pile Dataset) or bias-analysis work in the field. It might be useful to include a discussion of the relation of this work to the prior work in the next iteration of this submission.


**Summary And Contributions:**

This present paper provides a detailed discussion of the main characteristics of harmful text that are, or can be, generated by (large) language models. In Section 2, the authors outline six ways in which we can characterize harmful text generated by language models and discuss why such categorizations are needed and might be useful for us. One argument that the authors make is that such an abstract and methodological characterization of harmful text generated by language models would allow us to have a shared vocabulary and foundational language across studies that focus on biases contained in and harms inflicted by large language models. The authors apply their outlined harm characteristics to study the Perspective API and demonstrate a potential mismatch between the API’s original design principles and the API’s functional uses. Overall, the paper seeks to fill in an important gap in the study of harms and biases generated by (large) language models (and I applaud the authors for their work), but in my opinion, while the paper offers many critical insights and has many noteworthy merits; it might still benefit from a second round of reviews for the reasons that I will be discussing below. I am confident that the paper can be published at another great venue once the authors take the time to carefully address all the feedback and suggestions made by the reviewers.

**Note:** I am recommending a light rejection only because I believe the paper might benefit tremendously from going through another round of reviews to iron out some major issues surrounding Section 2 and Section 3.

---

> ### Author Response · Authors · 2022-08-24
> **Thank you, response and clarifications**
>
> We thank you for your detailed and constructive comments.  We are very pleased you believe “the paper seeks to fill in an important gap in the study of harms and biases” and applaud our work.  Below we address the critiques in the review.
>
> *“The authors fail to provide clear definitions of key terms that are used in the paper such as harm, bias, and toxic”*
> In an initial version, we included definitions of terms related to our characteristics but removed them due to lack of space.  Since we have an extra page for a revision, we have added these back in and hope they can help clarify our paper.
> In particular, we define harm as: “The real world effect on people that the evaluation’s metrics aim to approximate.”
> We appreciate the need for good definitions of terms like “bias” and “toxic”, however, we believe that there is not one correct definition of either of these terms.  For example, as is alluded to in L295-303 (original submission)/L322-332 (updated submission), what might be considered toxic in the context of an internet forum may not be toxic in a book.  As opposed to prescribing definitions for umbrella terms like bias and toxicity, we encourage curators of datasets and benchmarks to be explicit about the definition measured by their evaluation.  In the original submission this was referred to in L61-63 (L64-66 in the updated submission) and L331-333 (L332-334 in the update submission).  We have strengthened this point in Section 2.1 to make it clearer (L58-61 in the updated submission).
>
> *How does allocational harm differ from representational harm?*
> We hope the definitions added to Section 2.2 are helpful, but to expand on them: allocational harms occur when there is a material impact on a group of people.  For example, not receiving a loan due to gender is an example of a material impact on a person due to their identity.  In contrast, representational harms arise from (a tendency towards) portraying groups in a particular manner, leading to harmful downstream effects.  For example, portraying images of CEOs as consistently male does not create an immediate material discrepancy between genders but represents genders differently.  Representational harms could cause allocational harms “downstream” but representational harms can be particularly pernicious because the impact of such harms on individuals is more challenging to measure.  Please see Blodgett et al (“Language (Technology) is Power: A Critical Survey of “Bias” in NLP “) and Barocas et al (“The Problem With Bias: Allocative Versus Representational Harms in Machine Learning”) which introduce this distinction for more details.
>
> *The term “capability fairness” is truly confounding.*
> We hope the added definitions in section 2.2 are helpful.  Representational and allocational harms have been explored in the fairness literature, but when studying language models, we realized that many benchmarks do not fall under either representational or allocational.  In particular, consider one of our datasets in Table 1 (Deconfounding) which includes question/answer pairs and whether the pair pertains to a certain group (e.g., gender).  We can measure how well the model does when answering questions about certain groups, similar to how we could measure disparate outcomes (did an individual receive a loan or not?) when measuring allocational harms.  However, performance of a QA system usually does not directly impact material outcomes for users.  Consequently, Deconfounding can be measured as if it is an allocational harm but, like a representational harm, the material impact is unclear.  We believe that LM outputs can occupy a unique intermediate between representational and allocational harms.  This points to real research questions that benchmark designers should consider including: How do the harms measured by datasets like Deconfounding result in allocational (material) harm?  How do representational biases (e.g., associating different words with occupations) lead to capability fairness issues?

---

> > ### Author Response · Authors · 2022-08-24
> > **Thank you, response and clarifications (2)**
> >
> > *I really like the distinction between “instance” harm and “distributional” harm, but I believe this distinction was previously made*
> > Thank you; we also believe the distinction between instance and distributional harms is important.  We do not claim to be the first to note the difference between instance and distributional harms (we cite that Khalifa et al. makes a similar distinction and updated the paper to reflect that Gopher and PALM make this distinction).  Our contribution is highlighting how this distinction has bearing on benchmark design and contextualizing this distinction within the broader scope of benchmark design.
> >
> > We acknowledge that many of the points we make are similar to the Gopher paper as our analysis is heavily influenced by our experience applying benchmarks and evaluations to the Gopher model (L30).  We note that Gopher is not a published paper (nor is it currently in submission) and believe it should be treated as a preprint when assessing our work.
> >
> > Finally, though Perez et al. discuss instance and distributional harms, we see our work as complementary.  While Perez et al. elicit negative examples from models and use language against certain groups as one criteria to elicit bad language, we aim to clarify how the distinction between distributional and instance harms impacts benchmark design.  Our characteristics could be directly used to interrogate Perez et al.  For example:
> >
> > -- Are the comparison classes generated by a model meaningful (L172-179 in the original submission and L181-188 in the updated submission)?
> >
> > -- How are demographic groups marked in classes generated by a model (L167 in the original submission L176 in the updated submission)?
> >
> > -- Should templates generated by an LM be used to measure distributional or instance harms (L224-L226 in the original submission and L233-235 in the updated submission)? (e.g., one generated template is “How many GROUP people does it take to screw in a lightbulb?”; is it okay if the model answers in a derogatory way about a group as long as it does so for other groups, or should the model never respond in a derogatory way?)
> >
> > *I believe the contextual harms seem to contain a narrow and incomplete set of items. For instance, a humanist scholar (say, a sociologist) might find this categorization rather limited in its scope.*
> > Social context mentioned in the section would be reasonably expected to capture dimensions such as culture, socio-economic status, and linguistic norms (we explicitly mention cultural and language norms in L116 in the original submission and L123 in the updated submission).  Dialectal variation would be considered part of social context as well as we believe it sits at the intersection of socioeconomic and linguistic context.  As mentioned in line 1215 in the Appendix in the original submission and 1307 in the updated submission, we could even consider things like “temporality” to fall under social context.
> >
> > *The annotators, the curators, and the data owners, among some other groups.*
> > We agree that many people can be impacted when creating language harm benchmarks, including annotators, curators and data owners.  In our related work we cite some concurrent work that directly addresses some of these stakeholders (L345-346 in the original submission and L356-357 in the updated submission).  However, the primary objective of our characteristics is to help analyze how we measure the harm of LM generated text in benchmarks.  One could create a different set of characteristics (similar to what is done in Derczynski et al) for those involved in creating a dataset.  We believe it is useful to maintain separation between different characteristics to narrow the scope of any single piece of work and make thorough interrogation in a specific direction tractable.  We have added acknowledgement of this to our limitations section.

---

> > > ### Author Response · Authors · 2022-08-24
> > > **Thank you, response and clarifications (3)**
> > >
> > > *Justify or explain their characterizations in Table 1*
> > > Thank you for letting us know this was unclear; we have added this to our appendix and hope it is helpful to future readers.
> > >
> > > *Neither have they discussed how their work is related to other prior studies that have looked at the inherent (distributional) biases encoded in large-scale language modeling corpora*
> > > We do include a related work section, and note that we do relate our characteristics to related work in each section on characteristics.  In particular, as stated in L339 in the original submission and L249 of the updated submission, a plethora of recent work details different potential harms from LMs, but do not provide resources to help practitioners go from identifying a harm to measure to building a rigorous and reliable benchmark.  Some work touches on topics similar to our characteristics and we believe we have cited where we pull insight from this prior work throughout our paper (e.g., L225 of the original submission and L234 of the updated submission we cite insights that come from Stereotyping Norwegian Salmon: An Inventory of Pitfalls in Fairness Benchmark Datasets).  However, our work is an attempt to provide a comprehensive framework for practitioners and we believe it offers something unique to the current literature on how to measure harms in language generated by models.
> > >
> > > In reference to work looking at inherent biases in large-scale modeling corpora, we have added a citation to Dodge et al "Documenting Large Webtext Corpora:
> > > A Case Study on the Colossal Clean Crawled Corpus".  We believe understanding datasets is essential work and our characteristics could be used to design metrics for better dataset understanding.

---

> > > > ### Comment · Reviewer_6EL5 · 2022-08-29
> > > > **Score Change (From 5 to 7)**
> > > >
> > > > I thank and applaud the authors for taking the time to carefully address my and my fellow reviewers' comments and suggestions. I have changed my overall score from 5 to 7.

---

### Author Response · Authors · 2022-08-24
**Thank you for comments!**

We thank the reviewers for their constructive feedback.  Our paper presents six characteristics which can help guide practitioners when examining existing benchmarks and building new benchmarks to evaluate harm in language model outputs.  We are pleased that reviewers believed our work “seeks to fill in an important gap” (6EL5), that our “proposed characteristics are important and should be considered by future studies” (k8Xw), and “sheds light on distinctions that might be overlooked” (T9zc).

There were some concerns shared across reviewers that we address here, as well as in individual responses:

T9zc and M3My raised concerns that we did not include any new quantitative study or release a new benchmark.  We wanted to reiterate that we believe our work is in scope for the NeurIPS Benchmarks track which permits “**Frameworks for responsible dataset development**, audits of existing datasets, **identifying significant problems with existing datasets**”.  6EL5, M3My and 8rZR indicated that our work would be better if we included details on how we constructed the mapping in Table 1.  We have updated this in our Appendix (see Appendix B).  Additionally, we included more explicit definitions for our characteristics at the top of each subsection in section 2 to help with clarity critiqued by 6EL5.  Finally, we updated our revision to reflect many minor suggestions regarding grammar, clarity, and formatting.  Note our revision is now at 10 pages, in line with the revision guidelines we received from the program committee.

Please see individual responses for each reviewer for more details on our work.

---

### Meta-Review · Area_Chair_bgUm · 2022-09-09

**Recommendation:** Accept
**Confidence:** 4

**Metareview:**

The paper proposes a framework for characterizing harmful text generated from LLMs. This is a timely and extremely important topic, and the paper engages with the complex socio-technical questions around it. The reviewers are split in their opinions (three in favor of acceptance; three opposed). The key argument raised against the paper is that it does not introduce a new benchmark or dataset, and does not make falsifiable claims. The meta-reviewers consider the paper to be well in-scope for the track: despite lacking empirical results, it provides a "framework for responsible dataset development" and clearly identifies "significant problems with existing datasets."  The authors successfully address the other concerns. The paper makes an important contribution to the community and deserves to be accepted.

---

### Decision · Program_Chairs · 2022-09-16

Accept